# VLA-RFT: Vision-Language-Action Reinforcement Fine-Tuning with Verified Rewards in World Simulators

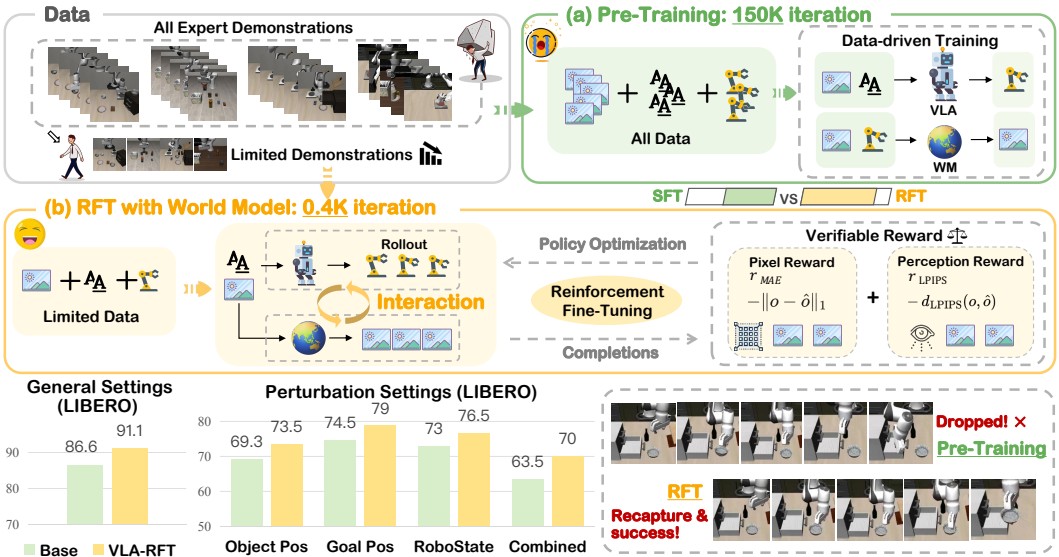

Figure 1: **The Framework of VLA-RFT.** A world model functions as a simulator that processes multi-rollout VLA action sequences to generate corresponding future states. By incorporating verified rewards through a GRPO optimization framework, we perform end-to-end updates of the VLA. Our approach achieves superior performance with remarkably fewer optimization steps—requiring only 0.4K iterations compared to 150K iterations for a strongly supervised baseline—demonstrating advantages in both standard and perturbed environments. Furthermore, the method exhibits enhanced execution-time robustness, characterized by reliable failure recovery and retry capabilities.

## Abstract

Vision-Language-Action (VLA) models enable embodied decision-making but rely heavily on imitation learning, leading to compounding errors and poor robustness under distribution shift. Reinforcement learning (RL) can mitigate these issues yet typically demands costly real-world interactions or suffers from sim-to-real gaps. We introduce VLA-RFT, a Reinforcement Fine-Tuning framework that leverages a data-driven world model as a controllable simulator. Trained from real interaction data, the simulator predicts future visual observations conditioned on actions, allowing policy rollouts with dense, trajectory-level rewards derived from goal-achieving references. This design delivers an efficient and action-aligned learning signal, drastically lowering sample requirements. With fewer than 400 fine-tuning steps, VLA-RFT surpasses strong supervised baselines and achieves greater efficiency than simulator-based RL. Moreover, it exhibits strong robustness under perturbed conditions, sustaining stable task execution. Our results establish world-model-based RFT as a practical post-training paradigm to enhance the generalization and robustness of VLA models.

# 1 INTRODUCTION

Vision-Language-Action (VLA) models have recently achieved remarkable progress by building upon large, pre-trained vision-language models (VLMs) (Li et al., 2025b; Karamcheti et al., 2024; Driess et al., 2023). Leveraging the powerful perceptual generalization of VLMs allows these models to operate under diverse visual conditions. However, most existing VLAs (Brohan et al., 2022; Zitkovich et al., 2023; Black et al., 2024; Bjorck et al., 2025; Kim et al., 2024) are trained purely via imitation learning. This approach is prone to error accumulation under distribution shift, where small deviations from expert demonstrations gradually drive the policy toward unfamiliar states and weaken its robustness (Ross & Bagnell, 2010; De Haan et al., 2019; Foster et al., 2024).

In contrast, reinforcement learning (RL) offers a promising avenue to overcome these limitations by explicitly optimizing beyond demonstrated behaviors and encouraging exploration (Liu et al., 2025). Recent studies have increasingly incorporated RL into VLA training, demonstrating its critical role in enhancing generalization and long-horizon task performance through offline RL approaches (Zhang et al., 2025c; 2024), direct real-world RL (Xu et al., 2024; Guo et al., 2025b), and simulation-based RL (Lu et al., 2025; Tan et al., 2025; Liu et al., 2025).

Yet, standard RL pipelines for VLA face steep challenges. Simulation-based RL (Chen & Li, 2025; Chen et al., 2025b; Shu et al., 2025) often requires millions of interactions and suffers from a pronounced sim-to-real gap. Real-world training (Xu et al., 2024; Mark et al., 2024; Guo et al., 2025c; Chen et al., 2025a), on the other hand, is prohibitively costly and can raise safety concerns. Offline RL also has inherent limitations: as noted by (Tan et al., 2025), without real environment interaction, models are vulnerable to distribution shift and cannot learn from the consequences of their own actions.

To address these challenges, we propose VLA-RFT, a Reinforcement Fine-Tuning framework that leverages a world model as a high-fidelity simulator for policy optimization. At its core, VLA-RFT employs a controllable world simulator that, once trained on a dataset of robot interactions, can predict future visual observations conditioned on an action sequence. Unlike conventional simulation environments restricted to handcrafted scenarios, this simulator is entirely data-driven, capturing the diversity of real-world interactions while avoiding the prohibitive cost and safety risks of training directly in the physical world. For a given task, policy-proposed actions are rolled out within this simulator to generate predicted visual trajectories. These synthetic trajectories then enable the design of a dense, task-grounded reward by comparing them against the visual trajectory from goal-achieving reference trajectory. These rewards are then used to optimize the policy via Generalized Reinforcement Policy Optimization (GRPO), enabling stable and efficient reinforcement fine-tuning.

This design provides a continuous, action-aligned learning signal that substantially reduces the sample complexity of reinforcement fine-tuning. Empirically, we show that with as few as 400 fine-tuning steps, VLA-RFT not only outperforms strong supervised fine-tuning baselines (Wang et al., 2025a) in both overall performance and compositional generalization, but also achieves markedly higher efficiency than simulator-based RL algorithms that demand orders of magnitude more interactions. Furthermore, in perturbed or adversarial scenarios, VLA-RFT exhibits superior action robustness, sustaining stable task execution even under unexpected environmental variations. Taken together, this combination of efficiency, generalization, and robustness underscores the practical advantages of our framework for scalable VLA training.

Finally, we hope that our method, experiments, and analysis will motivate future research to harness world models as a general and efficient post-training paradigm for VLAs, thereby substantially enhancing their practicality and accelerating their real-world deployment.

# 2 RELATED WORK

**Vision-Language-Action Models.** Vision-Language-Action (VLA) models align visual and linguistic inputs with actions through imitation learning on large-scale datasets (O'Neill et al., 2024; Liu et al., 2023; Mees et al., 2022). Pre-trained VLMs provide generalization, while supervised fine-tuning adapts them to task-specific action spaces (Li et al., 2025b; Karamcheti et al., 2024; Driess et al., 2023). Recent studies further improve efficiency with lightweight adapters and post-training techniques (Kim et al., 2025; Cui et al., 2025; Wang et al., 2025b; Fan et al., 2025; Gong et al.,

2024; Ding et al., 2024; 2025). However, imitation learning alone is prone to error accumulation under distribution shifts, where minor deviations from expert data push the policy into unfamiliar states and reduce robustness. To address this, recent studies incorporate reinforcement learning to improve VLA performance. Our work also falls into this line of research.

**VLA with Reinforcement Learning.** Reinforcement learning from human feedback has proven highly effective in language models (Sheng et al., 2024; Ouyang et al., 2022), inspiring RL fine-tuning for vision–language–action (VLA) systems. However, simulation-based RL (Chen & Li, 2025; Chen et al., 2025b; Shu et al., 2025) requires vast interactions and suffers from the sim-to-real gap, while real-world training (Xu et al., 2024; Mark et al., 2024; Guo et al., 2025c; Chen et al., 2025a) is expensive and unsafe. Offline RL also faces fundamental limitations: as highlighted by (Tan et al., 2025), policies struggle with distribution shift and the inability to learn from its own actions. To overcome these limits, we leverage a world model as a data-driven simulator, enabling practical policy optimization without real-world costs or risks.

**World Models and Verified Rewards** World Models learn environment dynamics for planning and control, either via explicit physics (Song et al., 2024; Li et al., 2024; Sancaktar et al., 2022) or latent predictive representations (Hafner et al., 2019b;a; 2023). Recent extensions integrate multi-modal inputs and guide RL with high-dimensional predictions (Wu et al., 2023; Li et al., 2025a). Advances in generative modeling (Ho et al., 2022; Blattmann et al., 2023; Liu et al., 2024) have enabled large-scale video-based World Models (Bardes et al., 2023; Assran et al., 2025), later specialized for robotics (Zhou et al., 2024a;b). Emerging works further link these models with instruction-conditioned action generation (Hu et al., 2024; Cen et al., 2025; Zhong et al., 2025; Zhang et al., 2025a). While these approaches explore diverse downstream applications, scaling World Models for VLA remains under-explored. VLA models—owing to high-dimensional visual and language inputs paired with fine-grained action outputs—require substantial data to scale effectively. Similar to the trajectory of LLM development, verified rewards (i.e., rewards that are deterministically computable and task-independent) are often more stable and reliable than learned reward models, which may suffer from task-specific overfitting, poor generalization, or reward hacking (Wen et al., 2025; Lambert et al., 2024; Guo et al., 2025a; Yue et al., 2025). Although some recent efforts explore training World Models using verifiable reward signals, none of these works leverage such rewards for reinforcement fine-tuning of VLA policies (Wu et al., 2025). In this work, our world model simultaneously acts as a dynamics simulator and as a source of verifiable reward signals for policy optimization, enabling reliable, fast, and scalable reinforcement fine-tuning of VLAs—without requiring human annotations, task-specific reward modeling, or online environment interaction.

## 3 METHOD

In this section, we begin by presenting the motivation behind our approach and outlining both the key challenges and the intuitive foundation of our pipeline. We then provide a formal problem definition and describe each component of the framework in detail. Finally, we present a comprehensive illustration of the two training phases, which is shown in Figure 2.

**Stage I: World Model (WM) and Policy Pretraining.** In the first stage, we pretrain the world model on offline datasets so that it can capture environment dynamics. In parallel, we pretrain the VLA policy to produce stable action chunks, which serve as a reliable initialization for subsequent optimization.

**Stage II: VLA Optimization through WM Interaction.** In the second stage, given an initial frame and a language instruction, the VLA rolls out $n$ action chunks. The world model then interactively generates trajectories conditioned on these actions and provides verified rewards. Using these feedback signals, the VLA is fine-tuned with GRPO to progressively improve policy performance.

### 3.1 PROBLEM FORMULATION

In this work, we investigate how to train a dual-system VLA policy equipped with a flow-matching action head, using both a WM and a verified reward mechanism. Specifically, we formulate the entire training process as a *Partially Observable Markov Decision Process (POMDP)*. The training pipeline is formally defined by the tuple

$$\mathcal{M} := (\mathcal{O}, \mathcal{S}, \mathcal{A}, \mathcal{L}).$$

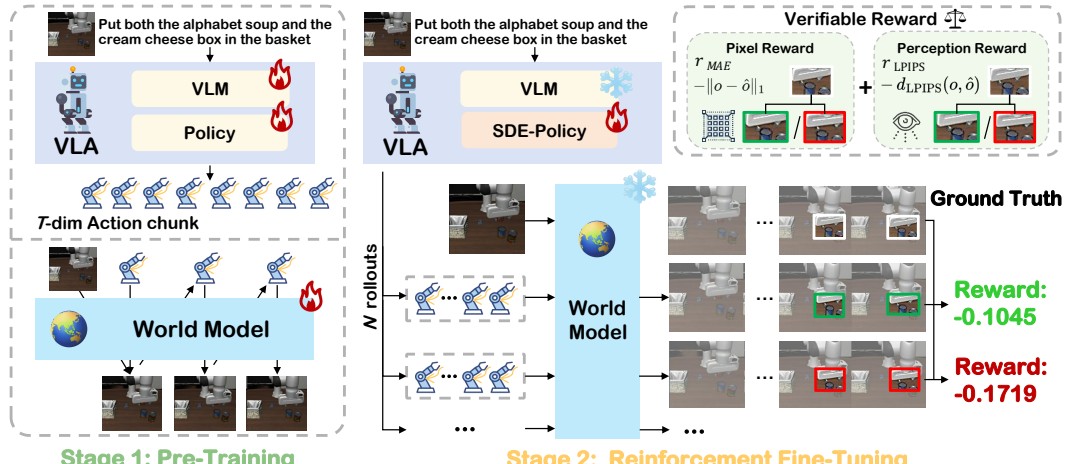

Figure 2: **Training Paradigm of VLA-RFT.** In the pre-training stage, both the world model and VLA policy are initialized, where the world model takes a 7-dimensional action input that is consistent in format with the VLA's action output. In the reinforcement fine-tuning stage, the VLA generates action chunks based on an initial frame and language instruction, which are rolled out in the world model to predict future states. Verified rewards are then computed from the predicted states and used to optimize the VLA via GRPO Optimization.

where Observations $\mathcal{O}$ represents the perceptual space of the agent, including real images captured from the environment. States $\mathcal{S}$ denotes the robot's proprioceptive state. Actions $\mathcal{A}$ is the action space. Language $\mathcal{L}$ refers to natural language instructions provided to the agent.

The VLA policy is expected to generate a sequence of $T$ actions with indices $t \in [T] = \{0, \ldots, T\}$, conditioned on the first observed real image $o_i$, the initial language instruction $l_i$, and the initial robot state $s_i$. This process is factorized as

$$\hat{a}_{i:i+T-1} \sim \pi_\theta\big(\cdot \mid o_i, l_i, s_i\big) = \pi_{\theta_{\mathrm{fm}}}\big(\cdot \mid z_i, s_i\big), \qquad z_i = f_{\mathrm{VLM}}(o_i, l_i). \tag{1}$$

where $f_{\mathrm{VLM}}$ denotes the vision–language large model that encodes multimodal inputs into latent representations $z_i$, and $\pi_{\theta_{\mathrm{fm}}}$ represents the flow-matching policy head that generates the corresponding action chunk.

The world model acts as an interactive simulator that generates rollouts conditioned on the first image $o_t$ and the policy-generated action sequence $a_{t:t+T-1}$. By comparing the generated trajectory against ground-truth images or ground-truth-action-induced rollouts, we obtain a verified reward signal:

$$\hat{o}_{i+t+1} = \begin{cases} g_\phi(o_i, a_i), & t = 0, \\ g_\phi(o_{i:i+t}, a_{i:i+t}), & t = 1, \ldots, T-1. \end{cases} \tag{2}$$

where $g_\phi$ denotes the autoregressive world model. In particular, the first prediction is generated from the initial frame $o_t$ and the first action $a_t$, while subsequent predictions ($i \geq 1$) are produced autoregressively by conditioning on both the previously generated frames $o_{t:t+i}$ and the executed actions $a_{t:t+i}$.

## 3.2 STAGE I: WM PRETRAINING AND VLA PRETRAINING

To reduce reinforcement learning instability and prevent early collapse, we pretrain the world model and policy on offline datasets, providing a stable initialization for subsequent optimization.

**World Model Training.** To obtain dense verified rewards more efficiently, and inspired by recent advances in video generation models (e.g., iVideoGPT (Wu et al., 2024)), we train an interactive video prediction model to serve as the world model. This design avoids the limitations of implicit world models, such as sparse reward signals and the lack of verifiable environment dynamics. It

consists of a pretrained tokenizer and an autoregressive Transformer backbone. During pretraining, the WM is optimized via maximum likelihood (MLE):

$$\mathcal{L}_{\text{MLE}}^{\text{WM}}(\phi) = -\mathbb{E}\Big[\log p_\phi(o_{i+1} \mid o_i, a_i) + \sum_{t=1}^{T-1} \log p_\phi(o_{i+t+1} \mid o_{i:i+t}, a_{i:i+t})\Big]. \tag{3}$$

where $p_\phi(\cdot)$ denotes the predictive distribution of future observations parameterized by the world model with parameters $\phi$.

**VLA Pretraining.** In this stage, we aim to ensure that the VLA produces stable actions. Since the flow-matching action head provides stable training for continuous actions, we pretrain the upstream VLM encoder and the flow-matching head on the expert demonstration dataset $\mathcal{D}$.

$$\mathcal{L}_{\text{MSE}}^{\text{VLA}}(\theta) = \mathbb{E}_{(a_{i:i+T-1},o_i,l_i,s_i)\sim D}\Big[\|\mathbf{v}_\theta(o_i, l_i, s_i, a_{i:i+T-1}^\tau) - u_\tau\|_2^2\Big]. \tag{4}$$

where $\tau \sim \text{Beta}(\alpha, \beta)$ is the flow-matching timestep, $v_\theta(\cdot)$ denotes the flow predicted by the action head parameterized by $\theta$, $a_{t:t+T-1}^\tau = \tau a_{t:t+T-1} + (1-\tau)\epsilon$ is the noise-perturbed action chunk, $u_\tau = a_{t:t+T-1} - \epsilon$ is the target flow field defined by the noisy action interpolation, and $\epsilon \sim \mathcal{N}(0, I)$ is standard Gaussian noise.

### 3.3 STAGE II: VLA OPTIMIZATION THROUGH WM INTERACTION

To achieve stable and efficient fine-tuning, we adopt an Stochastic Differential Equation (SDE)-based policy formulation optimized with GRPO, which offers reliable gradient estimates. The Stage I world model serves as an interactive simulator, providing verified rewards that further enhance training stability.

**SDE-Policy: Policy Parameterization via Flow and Sigma.** Since flow matching is inherently a deterministic Ordinary Differential Equation (ODE) process, it has limitations in directly obtaining log-likelihood. To address this, we build upon prior work on flow-matching reinforcement learning(e.g. ReinFlow (Zhang et al., 2025d)) by extending the framework into a stochastic formulation, thereby enabling exploration during training. In Stage II, we introduce a *Sigma Net*, whose architecture mirrors that of the flow-matching head, and which outputs a variance vector that parameterizes the stochasticity of the policy. Concretely, at inference time, we discretize the integration into $K = 10$ steps, with $k \in [0, 1, 2, \ldots, 10]$. Actions are generated by integrating the learned vector field from $\tau = 0$ to $\tau = 1$, initialized from random noise $a_{i:i+T-1}^{\tau=0} \sim \mathcal{N}(0, I)$. We apply the forward Euler method:

$$\mu_k = a_{i:i+T-1}^{k\delta} + \delta\mathbf{v}_\theta(o_i, l_i, s_i, a_{i:i+T-1}^{k\delta}), \tag{5}$$

where $\delta = 0.1$ is the integration step size. For each integration steps $k$, *Sigma Net* takes as input $(z_i, s_i, k)$ and outputs a variance vector $\sigma_\psi^k$, while the flow-matching action head simultaneously predicts the flow $\mu_k$. Together, these two components define a Gaussian conditional distribution from which the next action chunk is sampled, thereby generalizing the deterministic Flow Matching (FM)-ODE formulation into a SDE process:

$$a_{i:i+T-1}^{k\delta} \sim \mathcal{N}(\mu_k, \Sigma_k), \tag{6}$$

where

$$\Sigma_k = (\sigma_\psi^k)^2. \tag{7}$$

Within the same rollout, we compute the step-wise log-likelihoods across the $K$ denoising steps, and take their average as the log-probability of the rollout:

$$\bar{\ell}_{\theta,\psi} = \frac{1}{K} \sum_{k=1}^{K} \log p_{\theta,\psi}^{(k)}(a_{i:i+T-1}^{k\delta} \mid a_{i:i+T-1}^{(k-1)\delta}, z_i, s_i). \tag{8}$$

Finally, we compute the policy ratio with respect to the old policy by exponentiating the difference of average log-probabilities:

$$r = \exp(\bar{\ell}_{\theta,\psi} - \bar{\ell}_{\text{old}}). \tag{9}$$

**Interactive WM Simulation and Verified Reward.** Visual features often carry richer semantic information. To leverage this, given an action chunk $a_{t:t+T-1}^K$ from the SDE-Policy, the world

---

**Algorithm 1** VLA Fine-Tuning Pipeline with World Model and Verified Reward

---

**Require:** Offline dataset $\mathcal{D}$, diffusion horizon $K$, chunk length $T$, rollout number $N$, initial frame $o_t$, sigma net parameters $\psi$
**Ensure:** Trained VLA policy $\pi_\theta$
1: **Stage I: Pretraining**
2: Train WM parameters $\phi$ with maximum likelihood Eq. 3
3: Train VLA encoder $f_{\text{VLM}}$ + flow-matching head $\pi_{\theta_{\text{fm}}}$ with loss Eq. 4
4: **Stage II: Interaction and Optimization**
5: **for** each task instance **do**
6:     **for** $n = 1$ to $N$ **do**            ▷ *Rollouts*
7:         **for** $k = 1$ to $K$ **do**         ▷ *Diffusion steps*
8:            Sample actions from Gaussian distribution $p_{\theta,\psi}^{(k)}$    ▷ *Eq. 6*
9:            Calculate log-probability $\ell^{(k)}$           ▷ *Eq. 8*
10:        **end for**
11:        Generate trajectory Traj with WM        ▷ *Eq. 10*
12:        Compute verified reward $R_n$          ▷ *Eq. 11*
13:     **end for**
14:     Compute advantages $\text{Adv}_n = R_n - \bar{R}_{\text{group}}$
15:     Update policy $\pi_\theta$ and sigma net with GRPO objective    ▷ *Eq. 13*
16: **end for**

---

model generates a visual trajectory, which is aligned with ground-truth data to construct verified rewards. This design improves reward reliability, reduces manual labeling, and enhances stability.

Starting from the initial frame $o_i$ and the first action $a_i^K$, the WM generates the next frame and recursively conditions on previously generated frames to produce the complete trajectory:

$$\text{Traj} = \big[o_i,\, a_i^{K\delta},\, \hat{o}_{i+1},\, \ldots,\, a_{i+T-1}^{K\delta},\, \hat{o}_{i+T}\big], \tag{10}$$

The generated sequence $\hat{o}_{i+1:i+T+1}$ is aligned with the ground-truth frames $o_{i+1:i+T+1}$ from the offline dataset. The verified reward for the current trajectory segment is defined as the negative weighted sum of the per-frame reconstruction loss and perceptual similarity loss:

$$R = -\sum_{t=0}^{T-1} \Big[ \lambda_1\, L_1\big(\hat{o}_{i+t+1},\, o_{i+t+1}\big) + \lambda_{\text{lp}}\, \text{LPIPS}\big(\hat{o}_{i+t+1},\, o_{i+t+1}\big)\Big]. \tag{11}$$

To reduce variance, we group $n$ rollouts sampled from the same starting state and compute the group average reward as a baseline:

$$\bar{R}_{\text{group}} = \frac{1}{N}\sum_{j=1}^{N} R_j, \qquad \text{Adv}_n = R_n - \bar{R}_{\text{group}}. \tag{12}$$

Using the policy ratio $r$ derived earlier, the VLA policy is optimized with GRPO. For training stability, we also retain a small-weight flow-matching mean squared error (MSE) term as auxiliary supervision on the flow head. The final objective is

$$\mathcal{L}_{\text{GRPO}}^{\text{VLA}}(\theta, \psi) = -\mathbb{E}\big[\, \text{clip}(r,\, 1-\epsilon,\, 1+\epsilon)\, \text{Adv}\,\big] + \lambda_{\text{mse}}\, \mathcal{L}_{\text{MSE}}^{\text{VLA}}(\theta) - \alpha\, \mathbb{H}\big(\pi_{\theta,\psi}\big). \tag{13}$$

where $\mathcal{L}_{\text{MSE}}^{\text{VLA}}(\theta)$ is the auxiliary flow-matching MSE loss with weight $\lambda_{\text{mse}}$, and $\mathbb{H}(\pi_{\theta,\psi})$ is the policy entropy used to encourage exploration, weighted by $\alpha$. Therefore, the objective integrates policy optimization with auxiliary supervision to ensure efficient and stable fine-tuning.

## 4 EXPERIMENTS

In this section, we assess VLA-RFT through three research questions: 1) How well can world model approximate a simulator? 2) How does world model improve VLA performance? 3) Which components of VLA-RFT drive these improvements?

### 4.1 EXPERIMENTAL SETUP.

**Implementations.** **1) Benchmark**: We evaluate our model on the LIBERO benchmark (Liu et al., 2023). **2) Metrics**: We report Success Rate (SR) for all tasks. **3) Baseline**: To accelerate experimentation, we employed a lightweight variant of VLA-Adapter (Wang et al., 2025a) as our baseline. More details of policy choice can be found in Appendix A.1. **4) World Model**: To optimize the balance between training efficiency and generation quality, we implemented a lightweight autoregressive world model based on the LLaMA architecture (Touvron et al., 2023). This model was instantiated as a compact 138M-parameter variant, comparable in scale to GPT-2 small (Radford et al., 2019). The model underwent pretraining on the LIBERO dataset to effectively capture task-relevant visual and action dynamics. **5) Training Details**: We initially pretrained a initial policy through supervised fine-tuning. Subsequently, we conducted post-training with reinforcement fine-tuning (RFT) using VERL (Sheng et al., 2024), a distributed RL framework that coordinates diverse rollout strategies with FSDP-sharded training. All experiments were executed on 4× A800 GPUs.

### 4.2 WORLD MODEL CAPABILITIES.

**Experimental Setting.** To evaluate whether pre-training enables the world model to capture environmental dynamics, we assess its pixel-level generation capability. We randomly sample $T$ consecutive image-action pairs from LIBERO, input the initial frame and complete action sequence into the world model, and compare the generated frames with ground-truth images for subsequent steps.

**Results Analysis.** We partition the dataset into training and test splits at a 49:1 ratio, and report all evaluation results on the held-out test set. As shown in Table 1, the world model attains low reconstruction error (MSE 0.0039) and strong perceptual scores—PSNR (peak signal-to-noise ratio) of 25.23 dB, SSIM (structural similarity index) of 0.906, and LPIPS (Learned Perceptual Image Patch Similarity) (Zhang et al., 2018) of 0.059—indicating high frame fidelity and perceptual quality. Qualitative results show sharp, temporally consistent frames that capture both static backgrounds and action-driven changes, demonstrating that pre-training enables the model to learn visual appearance and action-conditioned dynamics.

Table 1: **World model generation performance.** Left: frame-level metrics across four suites (Spatial, Object, Goal, Long) and their averages—MSE (pixel error), PSNR (signal-to-noise ratio), SSIM (structural similarity), and LPIPS (perceptual distance). Right: qualitative results. Left column shows simulator sequences, right column shows world-model generations from the same initial frame and actions, illustrating consistent appearance and action-induced dynamics.

| Task | MSE ↓ | PSNR ↑ | SSIM ↑ | LPIPS ↓ |
|------|-------|--------|--------|---------|
| Spatial | 0.0039 | 24.98 | 0.896 | 0.067 |
| Object | 0.0036 | 25.13 | 0.913 | 0.054 |
| Goal | 0.0024 | 26.99 | 0.929 | 0.040 |
| Long | 0.0056 | 23.83 | 0.885 | 0.074 |
| Avg | 0.0039 | 25.23 | 0.906 | 0.059 |

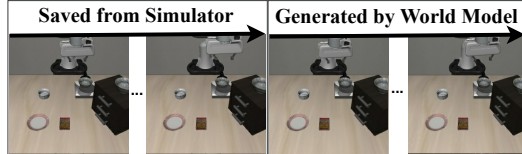

### 4.3 PERFORMANCE IMPROVEMENTS FOR VLA.

In the previous section, we analyzed the generation quality of the world model. Here, we further investigate whether our training pipeline enhances policy capability. Specifically, we evaluate policy performance before and after training under the following two task settings.

**LIBERO Standard Suites.** We evaluate RFT on the LIBERO Standard Suites using the Base model trained for 150k steps (Base-150k) as the baseline. As shown in Table 2, only 400 training steps of RFT (RFT-400) raise average SR from 86.6% to 91.3% (+4.7 points), with gains across all suites: Spatial (+6.0 points), Object (+6.4 points), Goal (+2.6 points), and Long (+3.0 points). The graph further shows RFT-400 consistently outperforms Base-150k. Notably, while extending supervised fine-tuning (SFT) training steps from 30k to 150k required heavy training, RFT delivers clear improvements with far fewer iterations, underscoring its efficiency.

**LIBERO Perturbation Suites.** To assess out-of-distribution robustness, we construct perturbed variants across the four LIBERO suites and report success rates for initial policy and our method.

Table 2: **Performance under LIBERO Standard Suites.** The table reports success rate (SR) across the four suites (Spatial, Object, Goal, and Long) and their average; the radar plot on the right provides a visual comparison of different model stages across tasks. Where "Base-30k" denotes a policy checkpoint after 30k steps of supervised fine-tuning (SFT), and "Base-150k" denotes a policy checkpoint after 150k SFT steps.

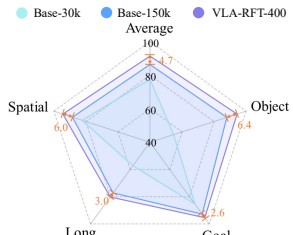

| Policy (iterations) | Spatial | Object | Goal | Long | Average |
|---|---|---|---|---|---|
| Base-30k | 82.4 | 84.8 | 85.4 | 57.2 | 77.5 |
| Base-150k | 88.4 | 88.0 | 92.8 | 77.2 | 86.6 |
| VLA-RFT-400 | **94.4** | **94.4** | **95.4** | **80.2** | **91.1** |
| Δ vs Base-150k | **+6.0** | **+6.4** | **+2.6** | **+3.0** | **+4.5** |

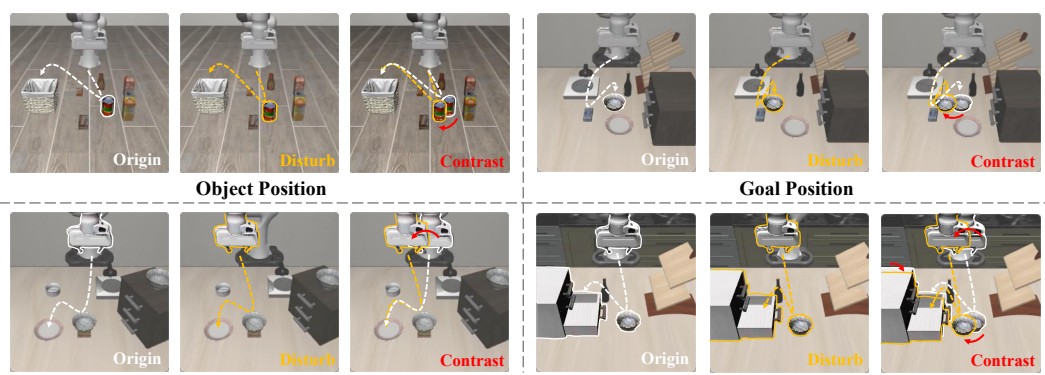

Figure 3: **Illustration of perturbed task settings in LIBERO.** We consider four perturbation types to evaluate out-of-distribution robustness: (Object Position) shifting the initial $(x, y)$ coordinates of the manipulated object; (Goal Position) displacing the target object in the $(x, y)$ plane; (Robot State) modifying the gripper's vertical height and horizontal offset; and (Combination) applying all perturbations together. Each row shows the original setting (Origin), the perturbed variant (Disturb), and a side-by-side comparison (Contrast).

**1) Experimental Setting.** In LIBERO-Object, the manipulated object's initial position is shifted in the $(x, y)$ plane with small or large offsets. In LIBERO-Goal, the target object's initial position is similarly displaced. In LIBERO-Spatial, the robot's initial state is perturbed by adjusting the gripper height and horizontal offset. In LIBERO-Long, we combine all the above perturbations. An illustration of the perturbed tasks is provided in Figure 3.

**2) Results Analysis.** As shown in Table 3, VLA-RFT consistently improves robustness across all types of perturbations. While Base-150k degrades substantially under larger shifts, VLA-RFT maintains higher stability, demonstrating its effectiveness against distributional shifts. The gains are most pronounced in the Goal and combined perturbations (over +6%), where generalization is more challenging, while RoboState perturbations show smaller but consistent improvements. Overall, our training pipeline not only increases standard performance but also improves out-of-distribution robustness, particularly in more complex settings. To further understand the robustness gains, we examine action distributions in Figure 4. VLA-RFT yields broader coverage

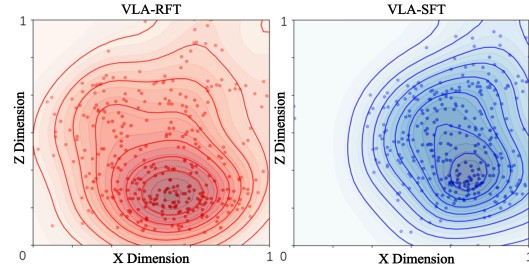

Figure 4: **Action distribution visualization of VLA-RFT and VLA-SFT.** The plots show distributions along $X$ and $Z$ action dimensions: the left plot corresponds to the RFT-trained policy, and the right plot to the SFT-only initial policy .

Table 3: **Performance under perturbation settings.** All perturbation magnitudes are in centimeter.

| Object Pos Perturb | Range | SR (%) |
|---|---|---|
| **Minor Perturbation** | | |
| Base-150k | ±2.5 | 69.3 |
| VLA-RFT | ±2.5 | 73.5 |
| Δ vs Base | ±2.5 | +4.2 |
| **Major Perturbation** | | |
| Base-150k | ±5 | 48.0 |
| VLA-RFT | ±5 | 52.5 |
| Δ vs Base | ±5 | +4.5 |

| Goal Pos Perturb | Range | SR (%) |
|---|---|---|
| **Minor Perturbation** | | |
| Base-150k | ±2.5 | 74.5 |
| VLA-RFT | ±2.5 | 79.0 |
| Δ vs Base | ±2.5 | +4.5 |
| **Major Perturbation** | | |
| Base-150k | ±5 | 44.8 |
| VLA-RFT | ±5 | 51.5 |
| Δ vs Base | ±5 | +6.7 |

| RoboState Perturb | Range | SR (%) |
|---|---|---|
| **Minor Perturbation** | | |
| Base-150k | ±20 | 73.0 |
| VLA-RFT | ±20 | 76.5 |
| Δ vs Base | ±20 | +2.5 |
| **Major Perturbation** | | |
| Base-150k | ±50 | 63.5 |
| VLA-RFT | ±50 | 67.0 |
| Δ vs Base | ±50 | +3.5 |

| Combined Perturb | Range | SR (%) |
|---|---|---|
| **Minor Perturbation** | | |
| Base-150k | ±2.5/2.5/20 | 63.5 |
| VLA-RFT | ±2.5/2.5/20 | 70.0 |
| Δ vs Base | ±2.5/2.5/20 | +6.5 |
| **Major Perturbation** | | |
| Base-150k | ±5/5/50 | 34.0 |
| VLA-RFT | ±5/5/50 | 37.0 |
| Δ vs Base | ±5/5/50 | +3.0 |

Table 4: **Reward design comparison on LIBERO.** The left table reports the average Success Rates (SR, %) of Base-150k and its variants trained with three different verified reward types. The right figure illustrates the corresponding reward function structures.

| Policy | Average (SR %) |
|---|---|
| **Base** | |
| Base (150k) | 86.6 |
| **Action Deviation Reward** | |
| VLA-RFT (R1) | 87.7 |
| Δ vs Base | +1.1 |
| **Image Consistency Reward** | |
| VLA-RFT (R2) | 87.1 |
| Δ vs Base | +0.5 |
| **Model-Based Image Consistency Reward** | |
| VLA-RFT (Ours) | **91.1** |
| Δ vs Base | **+4.5** |

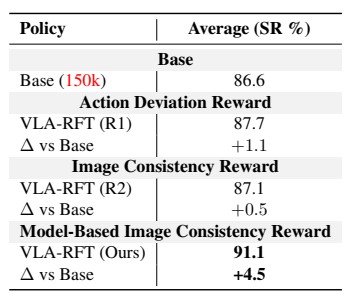 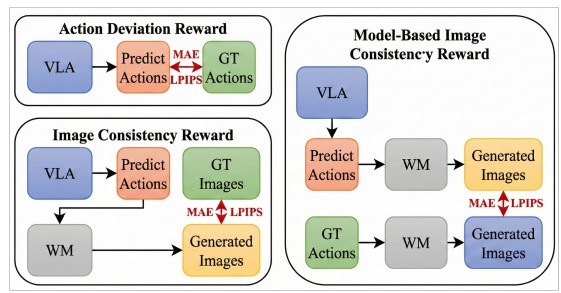

across action dimensions than initial policy, while SFT remains narrowly concentrated. This broader exploration enables better adaptability and generalization under perturbations.

## 4.4 KEY FACTORS FOR VLA-RFT

We showed our pipeline improves policy performance and robustness. Next, we test which components drive these gains via three verified reward designs and world model ablations.

**1) Experimental Setting.** We design three verified rewards under the same training setup and apply RFT to the base model to compare their effects on LIBERO success rates. "Action Deviation Reward" uses the negative L1 distance between policy and dataset actions, offering direct action-level supervision. "Image Consistency Reward" generates images from policy actions via the world model and compares them with dataset images using negative MAE and LPIPS, providing pixel-level guidance. "Model-Based Image Consistency Reward" renders trajectories from both policy and dataset actions within the same world model, using negative MAE and LPIPS across time to mitigate generation-quality bias and ensure fairness.

**2) Results Analysis.** As shown in Table 4, the comparison across reward designs highlights the essential role of the world model in the training pipeline. "Action Deviation Reward", which excludes the world model and relies only on action-level supervision, brings very limited gains (+1.1 points), showing that imitation alone is insufficient. "Image Consistency Reward" uses the world model and achieves moderate improvements, but direct comparison with real images still has limitations. "Model-Based Image Consistency Reward" maximally exploits the world model by performing trajectory comparisons within the same generative space, leading to consistent improvements across

all tasks and an average success rate of 91.1% (+4.5 points over the initial policy ). These results demonstrate that the world model is a key component, providing reliable optimization signals and enhancing both performance and robustness.

## 5 CONCLUSION & LIMITATION

In this work, we introduced VLA-RFT, a reinforcement fine-tuning framework that uses a learned world model as a controllable simulator. This approach enables efficient and safe policy optimization, bridges imitation and reinforcement learning, and reduces real-world interaction costs. Experiments show strong performance and generalization with minimal fine-tuning, highlighting world-model–based RFT as a promising direction for VLA research.

Nevertheless, several limitations remain. First, the verified reward is primarily based on the similarity between generated trajectories and expert demonstrations, constraining policies by dataset quality and limiting the discovery of strategies beyond expert performance. Second, the representational capacity of the world model remains a bottleneck; scaling to larger models trained on more diverse data could improve out-of-distribution generalization. Third, our framework does not explicitly integrate the world model into planning, which could enhance long-horizon reasoning. Finally, the reward mechanism itself could be improved—for example, by leveraging learned reward models (e.g., VLAC (Zhai et al., 2025)) to provide more task-relevant feedback. Extending the framework to a broader class of policy architectures also represents an important direction for future work.

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

# A APPENDIX

## A.1 MODEL ARCHITECTURE

**World Model.** As shown in Figure 5, given the input initial image, we first encode it using an encoder (similar to VQGAN (Esser et al., 2021)) to obtain image tokens, while continuous actions are discretized into action tokens through an action tokenizer. These image and action tokens are then jointly fed into the world model, which autoregressively predicts the future token sequences. Finally, the generated image tokens are decoded into corresponding future image sequences, enabling the modeling and simulation of environment dynamics. As shown in Table 5, the model is built on a 12-layer Transformer architecture with a hidden size of 768 and an intermediate FFN size of 3072. It employs 12 attention heads with a head dimension of 64, a maximum positional embedding length of 8192, SiLU activation, and a vocabulary size of 9008.

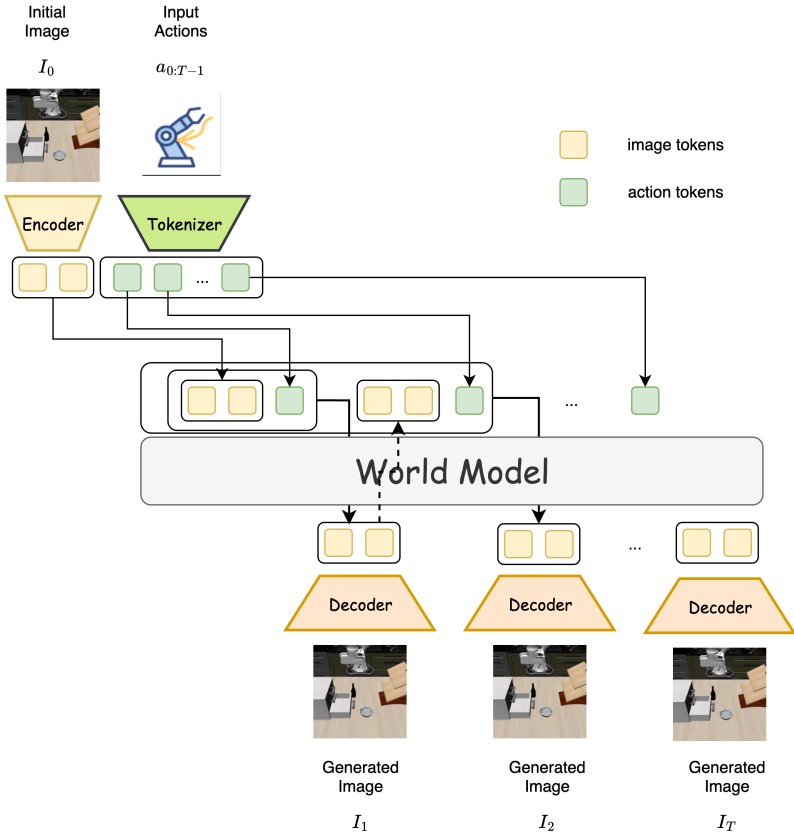

Figure 5: **Illustration of World Model Generation.** The initial image $I_0$ and input action sequence $a_{0:T-1}$ are first encoded into image and action tokens. These tokens are then fed into the world model to autoregressively predict the future state token sequence. Finally, decoders transform the generated image tokens into predicted future images $I_1, I_2, \ldots, I_T$.

**VLA Policy.** While flow-based methods such as $\pi_0$ (Black et al., 2024) demonstrate competitive performance, their JAX implementation poses integration challenges with VERL, and the LeRobot PyTorch version offers no significant advantages over VLA-Adapter despite its considerable computational overhead. Therefore, we selected VLA-Adapter (Wang et al., 2025a) as our base policy. During the RFT stage, we freeze the upper layer VLM of the policy and only update the lower layer action head. In addition, we incorporate a sigma net with a DiT-based architecture similar to the action head, which is responsible for generating noise outputs.

Table 5: Key hyperparameters of the World Model: Architecture (left) and Pre-training (right).

| Hyperparameter | Value |
|---|---|
| **Architecture** | |
| Layers | 12 |
| Hidden size | 768 |
| FFN intermediate size | 3072 |
| Attention heads | 12 |
| Head dimension | 64 |
| Key–value heads | 12 |
| Max position embeddings | 8192 |
| Activation | SiLU |
| Vocabulary size | 9008 |

| Hyperparameter | Value |
|---|---|
| **Pre-training** | |
| Training Steps | 100k |
| Batch size | 16 |
| Training steps | $1.5 \times 10^5$ |
| Learning rate | $5 \times 10^{-5}$ |
| Optimizer | AdamW (Kingma & Ba, 2014) |
| Datasets | Libero Datasets |
| Segment length | 8 |

Table 6: Key hyperparameters of the VLA-Adapter: Architecture (left) and Pre-training (right).

| Hyperparameter | Value |
|---|---|
| **Architecture** | |
| Vision backbone | dinosiglip-vit-so-224px |
| Input image size | $224 \times 224$ |
| LLM backbone | qwen25-0_5b-extra |
| LLM max length | 2048 |
| Text layers / hidden size | 24 / 896 |
| Attention heads / KV heads | 14 / 2 |
| FFN intermediate size | 4864 |
| Max position embeddings | 32768 |
| Torch dtype | bfloat16 |
| Action bins | 256 |

| Hyperparameter | Value |
|---|---|
| **Pre-training** | |
| Batch size | 16 |
| Training steps | $1.5 \times 10^5$ |
| Learning rate | $1 \times 10^{-4}$ |
| Optimizer | AdamW (Kingma & Ba, 2014) |
| Datasets | Libero Datasets |
| LoRA Rank | 64 |

## A.2 TRAINING DETAILS

**Pre-Training Phase.**

1) World Model: As shown in Table 5, the model is optimized using AdamW on the Libero datasets for $1.5 \times 10^5$ steps with a batch size of 16, a segment length of 8, and a learning rate of $5 \times 10^{-5}$.

2) VLA Policy: Our base policy consists of an upper-layer vision–language model (VLM) and a lower-layer DiT (Peebles & Xie, 2023)-based flow matching action head. During pre-training, we apply LoRA (Hu et al., 2022) for parameter-efficient fine-tuning of the VLM, while jointly optimizing the action head to better align the visual, language, and action spaces. The detailed architecture and training hyperparameters are summarized in Table 6.

**RFT Phase.**

For more details, see Figure 7.

1) World Model: The World Model is frozen.

2) VLA Policy: As shown in Table 7, we adopt GRPO (Chen et al., 2025b) as the advantage estimator and configure the optimization with a learning rate of $1 \times 10^{-6}$ and a sigma learning rate of $1 \times 10^{-5}$. For stability, an auxiliary MSE loss is included with coefficient 0.01, together with an entropy regularization term of 0.003 to encourage exploration. Training is conducted for 400 steps with a batch size of 16, and each update uses 16 rollouts. These settings strike a balance between stability and efficiency, enabling consistent improvements under limited compute budgets.

Table 7: Key hyperparameters for RL fine-tuning.

| Hyperparameter | Value |
|---|---|
| Advantage estimator | GRPO |
| Learning rate | $1 \times 10^{-6}$ |
| Sigma learning rate | $1 \times 10^{-5}$ |
| MSE loss coefficient | 0.01 |
| Entropy coefficient | 0.003 |
| Total training steps | 400 |
| Batch Size | 16 |
| Rollout Times | 16 |

Table 8: **Details of perturbation experiments.** Task 1 and Task 2 denote different tasks, while Dim 1 and Dim 2 refer to different perturbation objects or robot states. Where KP means keep original states.

| Policy. | Object Position | Goal Position | Robot Initial States | Task1 Dim1 SR (%) | Task1 Dim2 SR (%) | Task2 Dim1 SR (%) | Task2 Dim2 SR (%) | Average SR (%) |
|---|---|---|---|---|---|---|---|---|
| Base | ±2.5 | KP | KP | 87 | 52 | 78 | 60 | 69.3 |
| Ours | ±2.5 | KP | KP | 94 | 62 | 80 | 58 | 73.5 |
| Base | ±5 | KP | KP | 70 | 44 | 50 | 28 | 48.0 |
| Ours | ±5 | KP | KP | 72 | 52 | 56 | 30 | 52.5 |
| Base | KP | ±2.5 | KP | 62 | 58 | 92 | 86 | 74.5 |
| Ours | KP | ±2.5 | KP | 64 | 68 | 94 | 90 | 79.0 |
| Base | KP | ±5 | KP | 34 | 46 | 48 | 54 | 44.8 |
| Ours | KP | ±5 | KP | 46 | 42 | 58 | 60 | 51.5 |
| Base | KP | KP | ±20 | 60 | 88 | 54 | 90 | 73.0 |
| Ours | KP | KP | ±20 | 62 | 92 | 58 | 94 | 76.5 |
| Base | KP | KP | ±50 | 42 | 82 | 52 | 78 | 63.5 |
| Ours | KP | KP | ±50 | 46 | 86 | 56 | 80 | 67.0 |
| Base | ±2.5 | ±2.5 | ±20 | 64 | 82 | 36 | 72 | 63.5 |
| Ours | ±2.5 | ±2.5 | ±20 | 68 | 92 | 40 | 80 | 70.0 |
| Base | ±5 | ±5 | ±50 | 34 | 64 | 8 | 30 | 34.0 |
| Ours | ±5 | ±5 | ±50 | 36 | 60 | 12 | 40 | 37.0 |

## A.3 EXPERIMENT DETAILS

**Details of perturbation experiments.** The details of the perturbation experiments are shown in Table 8. Task 1 and Task 2 denote different tasks, while Dim 1 and Dim 2 refer to different perturbation objects or robot states.

**Comparisions with other VLA methods.** As shown in Table 9, VLA-RFT (Ours) consistently achieves the highest scores compared with baseline policies.

**Comparisons with other VLA+RL methods.** Our comprehensive evaluation demonstrates that the proposed framework achieves remarkable superiority over existing approaches across multiple dimensions. Not only does our method significantly outperform state-of-the-art offline RL baselines, but it also rivals the performance of online RL methods while maintaining the practical advantages of offline training. Most notably, our world-model-based approach delivers these superior results with dramatically reduced computational overhead, requiring substantially fewer training steps than conventional alternatives. The experimental comparison reveals the distinct advantages of our approach across diverse settings. While VLA-RL operates through direct reinforcement learning in the LIBERO environment, and competing methods like ARFM, RWR, and ReinboT represent the current best practices in offline RL, our framework consistently demonstrates superior performance gains. The key innovation lies in how VLA-RFT strategically exploits the world model's predictive capabilities to achieve unprecedented data efficiency, enabling faster convergence without sacrificing

Table 9: **Performance under general settings of LIBERO suites.** We report SR (%) across the four suites (Spatial, Object, Goal, and Long) and their average. VLA-RFT (ours) consistently achieves the highest scores compared with baseline policies. VLA-Adapter (Base) is the recurrence result when the Policy is Flow-matching and there is only one image input.

| Policy | Spatial | | Object | | Goal | | Long | | Average | |
|---|---|---|---|---|---|---|---|---|---|---|
| | SR (%) | Rank | SR (%) | Rank | SR (%) | Rank | SR (%) | Rank | SR (%) | Rank |
| Diffusion Policy (Chi et al., 2023) | 78.3 | 11 | 92.5 | 5 | 68.3 | 11 | 50.5 | 11 | 72.4 | 11 |
| Octo (Ghosh et al., 2024) | 78.9 | 9 | 85.7 | 10 | 84.6 | 5 | 51.1 | 10 | 75.1 | 9 |
| MDT (Reuss et al., 2024) | 78.5 | 10 | 87.5 | 9 | 73.5 | 10 | 64.8 | 5 | 76.1 | 8 |
| OpenVLA (Kim et al., 2024) | 84.7 | 7 | 88.4 | 8 | 79.2 | 7 | 53.7 | 9 | 76.5 | 7 |
| SpatialVLA (Qu et al., 2025) | 88.2 | 4 | 89.9 | 7 | 78.6 | 8 | 55.5 | 7 | 78.1 | 6 |
| WorldVLA (Cen et al., 2025) | 87.6 | 5 | 96.2 | 2 | 83.4 | 6 | 60.0 | 6 | 81.8 | 4 |
| CoT-VLA (Zhao et al., 2025) | 87.5 | 6 | 91.6 | 6 | 87.6 | 4 | 69.0 | 4 | 81.1 | 5 |
| TraceVLA (Zheng et al., 2025) | 84.6 | 8 | 85.2 | 11 | 75.1 | 9 | 54.1 | 8 | 74.8 | 10 |
| $\pi_0$ (Black et al., 2024) | 91.2 | 2 | 93.2 | 3 | 93.8 | 2 | 74.2 | 3 | 88.1 | 2 |
| VLA-Adapter (Wang et al., 2025a) (Base) | 88.4 | 3 | 92.8 | 4 | 88.0 | 3 | 77.2 | 2 | 86.6 | 3 |
| VLA-RFT (Ours) | **94.4** | 1 | **94.4** | 1 | **95.4** | 1 | **80.2** | 1 | **91.1** | 1 |

performance quality. For transparency and reproducibility, we note that VLA-RL results are sourced directly from the original publication, while the performance metrics for ARFM, RWR, and ReinboT on LIBERO are derived from the ARFM paper, ensuring fair and comprehensive benchmarking across all methods.

Table 10: **Comparison with other RL methods on Libero Average.** We report baseline success rate (SR), fine-tuned SR, their improvement ($\Delta$), and training steps.

| Type | Algorithm | Baseline SR (%) | SR (%) | $\Delta$ SR (%) | Training Steps |
|---|---|---|---|---|---|
| Online | VLA-RL (Lu et al., 2025) | 76.5 | 81.0 | **4.5** | 10,000 |
| | RIPT-VLA (Tan et al., 2025) | 96.7 | 97.5 | 0.8 | - |
| Offline | ARFM (Zhang et al., 2025b) | **88.1** | **92.1** | 4.0 | 40,000 |
| | RWR (Peters & Schaal, 2007) | **88.1** | 90.8 | 2.7 | 40,000 |
| | ReinboT (Zhang et al., 2025c) | **88.1** | 91.2 | 3.1 | 40,000 |
| Ours | **VLA-RFT** | 86.6 | 91.1 | **4.5** | **400** |

**Visualization.** We also provide more detailed visualization results in Figure 6 and Figure 8.

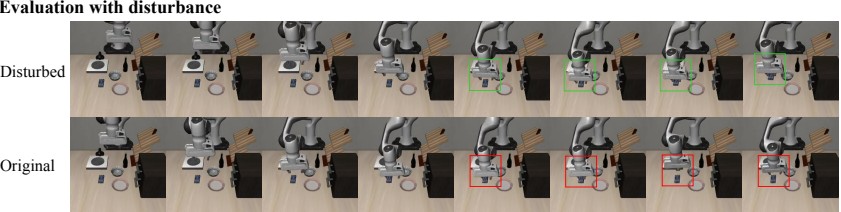

Figure 6: **Comparison of original and disturbed scenarios.**

## A.4 THE USE OF LARGE LANGUAGE MODELS (LLMS)

To enhance the readability and coherence of this paper, we employed large language models to assist in refining the writing.

## A.5 REAL WORLD EXPERIMENTS

**Experimental Setup. 1)Hardware Configuration:** We conduct our real-world experiments on a unified robotic platform. The system comprises a Flexiv Rizon 4s, a 7-DoF adaptive robotic arm known for its precise force control, equipped with a Flexiv GN01 two-finger gripper as the end-effector. For visual perception, we employ a single Intel RealSense D435i RGB-D camera mounted in a fixed third-person view. This camera setup provides a global perspective of the workspace, capturing RGB images necessary for the policy inputs. The entire system is powered by a workstation

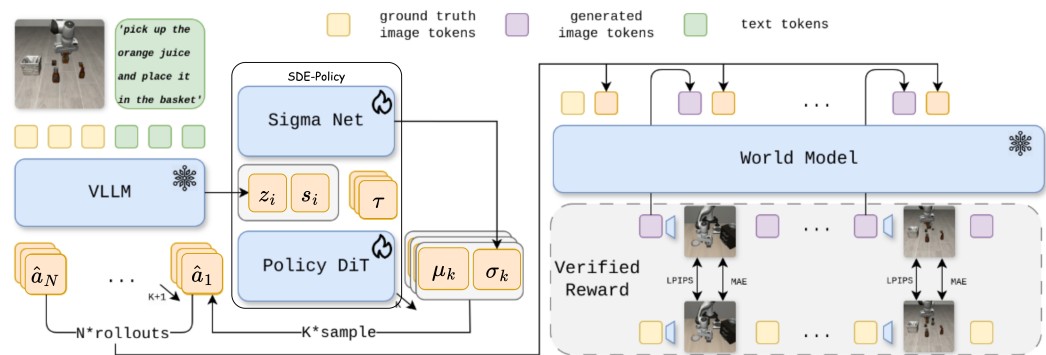

Figure 7: **Detailed Implementation of Method.**

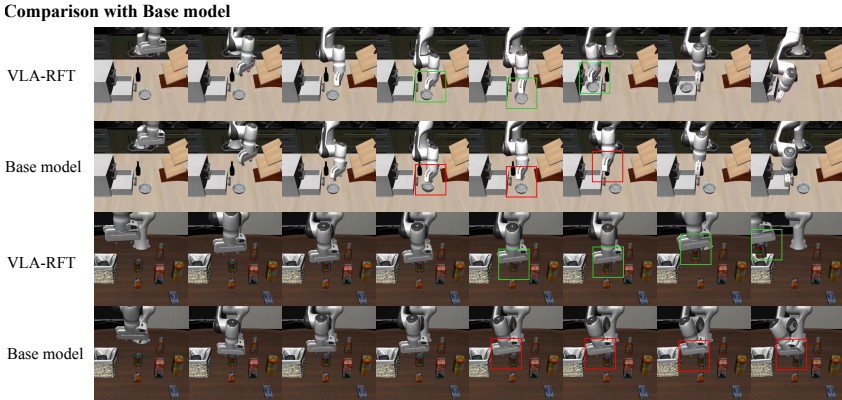

Figure 8: **Comparison of base policy and VLA-RFT.**

equipped with an NVIDIA RTX 4090 GPU to ensure real-time inference. **2)Task Definition:** We focus on the challenging task of cloth manipulation, specifically *Towel Folding*. The objective is to transform a towel from an initial flat state into a specific folded configuration. Due to the highly deformable nature of the fabric, this requires the agent to perform precise pick-and-place actions and dynamic adjustments.Each evaluation episode begins with the towel placed within the robot's workspace. The agent is tasked with completing the folding procedure within a strict time limit of 3 minutes. An episode is considered successful only if the towel is folded into the target structure and neatly organized within this duration.

**Training Details.** Initially, we collected a dataset consisting of 50 expert demonstration episodes. Using this dataset, we pre-trained the flow-based VLA-adapter policy for 20k and 80k steps, and the world model for 24k steps, all with a batch size of 16. Subsequently, we fine-tuned the policy for an additional 200 steps using our proposed RFT paradigm. All training procedures were conducted on a server equipped with eight NVIDIA H200 GPUs.

**Result Analysis.** We conducted a comparative evaluation between the policy checkpoint obtained after 20k steps of Supervised Fine-Tuning (SFT) and the checkpoint derived from the subsequent 200 steps of Reinforcement Fine-Tuning (RFT). The quantitative results are presented in Table 11. We observe that simply extending SFT by an additional 60k steps yields no further improvement in the success rate. In contrast, applying our RFT on top of the 20k-SFT checkpoint significantly boosts the success rate to 100% (10/10). This improvement is particularly pronounced in mitigating specific failure modes, such as unsuccessful grasping or premature dropping of the towel during transport.

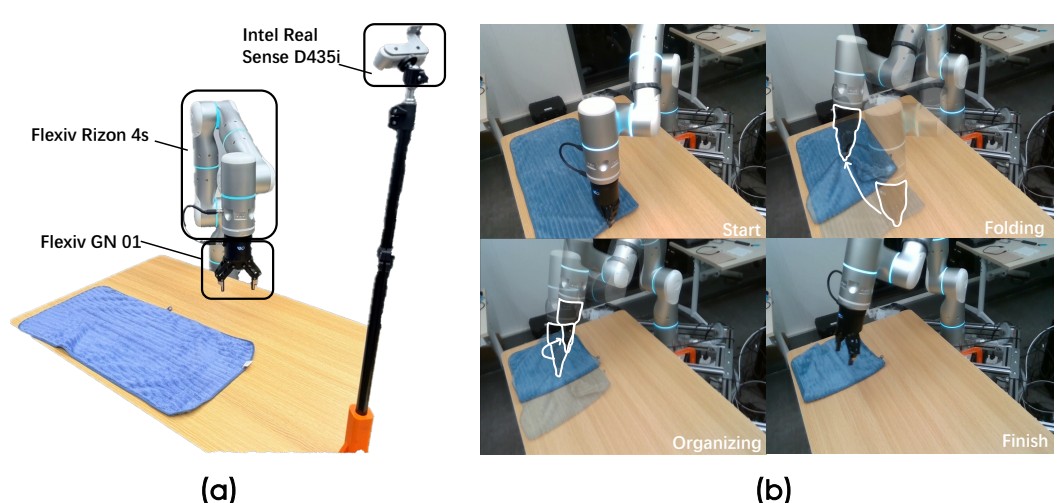

(a)  (b)

Figure 9: **Real World Experiments. (a)** The hardware platform setup used for data collection and policy evaluation. **(b)** A representative execution sequence of the towel folding task.

Table 11: **Real-world Experiment Results.** We report the success rate (SR) and detailed outcomes for 10 consecutive trials. For successful trials, the completion time in **seconds (s)** is recorded. Failure modes are noted explicitly.

| Trial ID | Base (20k SFT) | Base (80k SFT) | RFT (Ours) (20k SFT + 200 RFT) |
|---|---|---|---|
| 1 | *poor grasp* | *poor grasp* | **41** |
| 2 | 60 | **48** | 54 |
| 3 | *no grasp* | *poor grasp* | **60** |
| 4 | 56 | **42** | 54 |
| 5 | *no grasp* | **50** | 56 |
| 6 | *joint limit* | *joint limit* | **50** |
| 7 | *no grasp* | *no grasp* | **52** |
| 8 | 55 | **48** | 50 |
| 9 | **52** | *no grasp* | 60 |
| 10 | 51 | **49** | 62 |
| **SR** | 5/10 | 5/10 | **10/10** |

