# OpenReview forum: "VLA-RFT: Vision-Language-Action Reinforcement Fine-Tuning with Verified Rewards in World Simulators"
_ICLR.cc/2026/Conference — Submitted to ICLR 2026_

### Official Review · Reviewer_p7T1 · 2025-10-29

**Soundness:** 3
**Presentation:** 3
**Contribution:** 3
**Rating:** 6
**Confidence:** 4

**Summary:**

In this paper, the authors propose VLA-RFT, an enhanced fine-tuning framework based on the World Model, to improve the robustness and generalization ability of the vision-language-action model under distributional shifts. The core idea is to use a world model trained from real interactive data to generate future visual states as a high-fidelity simulator in which the rollout of policy actions is performed, the verified reward is constructed by the pixel-level and perception-level similarity with the expert trajectory (such as L 1 + LPIPS), and then the VLA strategy is efficiently fine-tuned by the GRPO algorithm. Experiments show that VLA-RFT can significantly outperform the strongly supervised fine-tuning baseline (+4.5% average success rate) on the LIBERO benchmark with only 400 fine-tuning iterations and exhibits stronger robustness across multiple perturbation settings.

**Strengths:**

- This paper accurately points out that the current mainstream VLA model relies on Imitation Learning, which leads to the problem of error accumulation and weak generalization ability outside the distribution. The limitations of traditional reinforcement learning (such as simulation-based training or real-world interaction) in terms of sample efficiency, security, and the sim-to-real gap are demonstrated.
- The data-driven world model is used as a verifiable reward generator instead of a traditional physical simulator, which skillfully avoids the problems of manual modeling bias and high cost. By calculating dense rewards between the generated and expert trajectories, a stable, aligned action-monitoring signal is provided for policy optimization, significantly reducing sample complexity.
- It surpasses the baseline requiring 150,000 steps of supervised training with only 400 fine-tuning steps and outperforms other RL methods requiring tens of thousands of steps (e.g., ARFM, ReinboT), highlighting the potential of the framework in practical deployment.

**Weaknesses:**

- Current reward mechanisms are entirely based on the similarity of the generative trajectory to the expert's trajectory, which means that the strategy can not surpass the expert's performance, nor can it explore better but look different solutions. If the expert data itself is suboptimal or biased (e. g. only one way of fetching) , the model will be limited to this range. It is suggested to discuss or experimentally verify the performance under non-optimal expert data.
- L1 + LPIPS, while a measure of visual fidelity, is not necessarily strongly associated with task success (e.g. , slight object rotation may not affect the task but result in high LPIPS) . If we can introduce task-related semantic rewards (such as critic model in VLAC) or learnable rewards, the alignment can be further improved.
- Although compared with ARFM, ReinboT, etc. , it does not cover areas such as Zhang et al. . (2025d) offline RL or Tan et al. . (2025) interactive post-training and other recent work. In addition, it is not stated whether attempts have been made to fine-tune (even on a small scale) the sim-to-real gap directly in the real world to see if it is truly eliminated.

**Questions:**

- If there are multiple ways to succeed in an expert trajectory (such as grabbing from the left or right), does the current reward based on a single reference trajectory penalize other effective strategies? Are multiple reference trajectories or post-clustering trajectory prototypes considered to construct rewards?
- Are reward signals misleading when the trajectories generated by the world model may be distorted in actions or scenarios not covered by the training data? Are there mechanisms to detect or suppress the effects of low-confidence rollouts (e.g., based on generation uncertainty weighting)?
- While RFT requires only 400 steps of policy updating, World Model pre-training takes 150,000 steps. Should the training cost of the world model be included in the overall efficiency assessment? The marginal cost of the world model is lower if it can be reused for multiple tasks, but this should be made clear.

---

> ### Author Response · Authors · 2025-11-26
> **5. Response to the Reviewer p7T1 (part1)**
>
> We thank the reviewer for the thorough and constructive feedback. We are glad that you find the problem formulation, the world-model-based reward design, and the empirical efficiency and robustness gains to be valuable. Below we address the raised concerns and questions in detail.
>
> #### 5.1. Limitation of expert-similarity rewards (cannot surpass expert / biased demos)
>
> First, we would like to reiterate that the challenges associated with using **ground-truth action-induced signals** are *not* incidental design flaws, but are instead inherent to the overarching goal articulated in our **“General Response to All Reviewers”**—namely, the pursuit of a **third, world-model-centered paradigm for solving real-world RL**.
>
> We are fully aware of the limitations of this design, and we explicitly discuss them in the *Limitations* section. In our view, this is a **practical trade-off that any current attempt to use world models for real-robot RL must confront** [2,3]: in many realistic settings, we simply do not have direct access to clean, simulator-style rewards.
>
> Therefore, our method is **not** proposed as an idealized solution for “how to optimize VLA policies with a world model under perfect reward supervision.” Instead, it is intended as a **temporary but practical and effective compromise** for scenarios where *reliable rewards are unavailable* and one still wishes to leverage a world model for real-robot RL.
>
> With this premise clarified, we now respond to your questions in detail:
>
> When the environment’s reward signal is not available to us, there are two straightforward options:
>
> - (a) **Learn a reward model**, as in [4, 5];
> - (b) **Use ground-truth subgoal images to compute a verified reward**, as in [1, 2, 3].
>
> For (a), designing a reward function/model is sometimes unreliable. Many recent approaches that attach a reward head to a VLM, or directly use a VLM as a verifier, are **highly dependent on the VLM’s understanding capability**. However, current VLMs are primarily trained for generic vision–language tasks and remain relatively weak in **robotics-specific spatial and dynamical reasoning**. Moreover, even for those works that perform task-specific finetuning, the underlying VLM models are typically not large enough to close this gap. As a result, the capability of current VLMs is still insufficient, and reward models built on top of them are prone to **reward hacking**.
>
> Therefore in this paper, we adopt **option (b)**. For real-robot RL at the current stage, **task success rate** is often more important than whether the explored trajectory is the shortest or fastest possible. Under this perspective, using ground-truth action-induced trajectories is a simple yet effective way to make real-robot RL actually work in practice. This choice is also consistent with many recent works (e.g., SRPO [3], NORA-1.5 [2]), which optimize policies with respect to goal images or latent representations (e.g., JEPA features) derived from expert behaviors rather than from truly optimal trajectories.
>
> Moreover, recent works such as **VLA-R1** [6] and **ThinkAct** [7] further support the idea that **consistency in pixel space** is often the most direct indicator of task execution quality. The main open question is not whether pixel space is appropriate, but rather **which pixel-space metric** to use—this is ultimately a modeling and design choice.
>
> [1] ReinboT: Amplifying Robot Visual-Language Manipulation with Reinforcement Learning. ICML2025
>
> [2] NORA-1.5: A Vision-Language-Action Model Trained using World Model- and Action-based Preference Rewards, arXiv (Nov 18)
> [3] SRPO: Self-Referential Policy Optimization for Vision-Language-Action Models, arXiv (Nov 19)
>
> [4] π\*0.6 : a VLA That Learns From Experience
>
> [5] World-Env: Leveraging World Model as a Virtual Environment for VLA Post-Training
>
> [6] VLA-R1: Enhancing Reasoning in Vision-Language-Action Models
>
> [7] ThinkAct: Vision-Language-Action Reasoning via Reinforced Visual Latent Planning
>
> From our perspective, these are **different solution routes**, rather than strictly ordered, superior–inferior options. Using ground-truth action-induced trajectories is a feasible, though not ultimately optimal, approach at the current stage, and we look forward to more sophisticated solutions that can address these challenges in the future.
>
> ####

---

> ### Author Response · Authors · 2025-11-26
> **5. Response to the Reviewer p7T1 (part2)**
>
> #### 	 5.2. L1 + LPIPS and its semantic alignment with task success
>
> We acknowledge that our definition of the reward signal has certain limitations.
>
> Although Section 5.1 explains why this design is reasonable and can work to some extent, there is indeed substantial room for improvement in the reward formulation.
>
> For example, following Nora 1.5 [2], one could incorporate features from VJEPA-2 [8], or use the VLAC [8] features you suggested, which would very likely improve the accuracy of the RL optimization process. Due to time constraints (most of our effort was devoted to real-robot experiments), we were not able to explore and discuss this direction in depth in the current version. However, we commit to incorporating such improvements in the final version.
>
> [8] VLAC: vision-language-action-critic model for robotic real-world reinforcement learning
>
> [9] V-JEPA 2: Self-Supervised Video Models Enable Understanding, Prediction and Planning
>
> #### 	      5.3. Recent related work and sim-to-real discussion
>
> ##### 5.3.1 About the related work
>
> It appears that Zhang et al. (2025d) actually corresponds to the same line of work as AFRM, which we have already compared against in Table.10. Regarding Tan et al. (2025), we have discussed it in the Introduction and Related Work sections.
>
> Regarding **RIPT-VLA** (Tan et al., 2025), we did not include a comparison in the current version because we were unable to identify an appropriate training-steps setting at the time. Therefore, we only reported results for VLA-RL. Following your suggestion, we will add RIPT-VLA as well. We will also incorporate more related baselines in the final version of the paper.
>
> ##### 5.3.2 About the sim-to-real discussion
>
> First, in **“General Response to All Reviewers”**, we have illustrated our goal is *not* to develop yet another simulation-based method followed by a sim-to-real transfer. Instead, we aim to pursue a **third, world-model-centered paradigm for solving real-world RL**.
>
> Due to time constraints at the submission deadline, we were indeed unable to include real-robot experiments, and we focused on simulation-only results, which already contained many technical contributions sufficient to validate our ideas.
>
> However, we have now added real-robot experiments. Surprisingly, on the *fold the towel* task—where the simulator cannot faithfully model the environment—our RFT approach achieves a **much larger improvement over SFT**. In particular, at the critical moments of grasping a deformable object, the SFT-only policy often fails to grasp the towel reliably because of its highly variable shape, leading to a significantly reduced success rate. In contrast, after RFT, the policy becomes much more precise at grasping such deformable objects and reaches a **100% success rate** on this task.
>
> Although these experiments are still preliminary, they already demonstrate that in scenarios where simulation cannot faithfully model the environment for RL optimization, our paradigm substantially advances a **third, real-world–oriented RL paradigm** based on world models.
>
> | Real world | Iter                  | SR    | Time(1)    | 2    | 3          | 4    | 5        | 6           | 7        | 8    | 9        | 10   |
> | ---------- | --------------------- | ----- | ---------- | ---- | ---------- | ---- | -------- | ----------- | -------- | ---- | -------- | ---- |
> | BasePolicy | 20k (SFT)             | 05/10 | poor grasp | 1'00 | no grasp   | 0'56 | no grasp | joint limit | no grasp | 0'55 | 0'52     | 0'51 |
> | BasePolicy | 80k (SFT)             | 05/10 | poor grasp | 0'48 | poor grasp | 0'42 | 0'50     | joint limit | no grasp | 0'48 | no grasp | 0'49 |
> | RFT        | 20k (SFT) + 200 (RFT) | 10/10 | 0'41       | 0'54 | 1'00       | 0'54 | 0'56     | 0'50        | 0'52     | 0'50 | 1'00     | 1'02 |
>
> ####

---

> ### Author Response · Authors · 2025-11-26
> **5. Response to the Reviewer p7T1 (part3)**
>
> #### 5.4 Other Questions:
> ##### Q1. Multiple successful strategies vs. single reference trajectory
>
> Our reward encourages the policy to stay close to the modes present in the expert dataset: any strategy that appears in the demonstrations is naturally rewarded. In other words, “multiple reference trajectories” already exist implicitly in the dataset. As long as both “left-grasp” and “right-grasp” behaviors appear in demonstrations, their corresponding trajectories are treated as valid success modes and are rewarded accordingly.
>
> The issue you refer to—multiple successful modes in expert data (e.g., grasping from the left or right)—is fundamentally a multi-modality problem. Importantly, this challenge is not specific to RL fine-tuning; it already exists in the SFT/BC pretraining stage. In fact, the problem is often more severe during pretraining because the expert data is noisier and multimodal behaviors are mixed without structure.
>
> Recent VLA works (e.g., the Pi-series models [2]) have shown that flow matching is highly effective for modeling multi-modal action distributions. Flow matching avoids collapsing modes and can faithfully model diverse behaviors in the expert dataset. Consistent with these findings, our RFT framework adopts flow matching to train the action expert, which effectively alleviates mode-collapse issues and preserves multi-modality during fine-tuning.
>
> Therefore, while our reward is based on pixel consistency with expert rollouts, it does not restrict the policy to a single canonical trajectory. Instead, combined with flow matching, it allows the policy to follow any of the valid expert-supported modes without penalizing legitimate alternative strategies.
>
> ##### Q2. Reward reliability under world model errors / OOD states
>
> We fully acknowledge the potential OOD issues in world-model rollouts, and our method is designed specifically to mitigate these effects.
>
> First, our reward formulation explicitly penalizes behaviors that deviate from the data distribution. When the policy proposes an action outside the training manifold, the world model is more likely to generate distorted or low-quality predictions. In such cases, the predicted image will naturally have lower similarity to the ground-truth-action–induced image, resulting in a lower reward. Thus, OOD rollouts do not produce misleadingly high reward; instead, they are penalized by design.
>
> Second, to further avoid error accumulation, our policy is never conditioned on world-model rollouts. All policy inputs come directly from the dataset rather than recursively generated states. This prevents the classic compounding-error problem and avoids drifting into regions where the world model is unreliable. In other words, the world model is used only for reward computation, not as a closed-loop simulator.
>
> Therefore, even when the world model encounters OOD states, the framework remains robust.
>
> OOD actions naturally lead to low similarity scores,and distorted rollouts are automatically suppressed via the reward mechanism. This ensures that the reward signal does not mislead the policy and instead encourages the model to remain in high-confidence regions supported by the dataset. We are also aware of this issue. Therefore, when designing our reward, we followed the robustness-enhancing strategy in GEVRM [1], which addresses inaccuracies in world model predictions. Specifically, we adopt the "Model-Based Image Consistency Reward " reward design, a typical “IMC”-style approach, to effectively improve robustness against world model errors.
>
> [1] GEVRM: Goal-Expressive Video Generation Model for Robust Visual Manipulation，ICLR25.
> [2] π0: A Vision-Language-Action Flow Model for General Robot Control

---

> ### Author Response · Authors · 2025-11-26
> **5. Response to the Reviewer p7T1 (part4)**
>
> ##### Q3. Accounting for world model pre-training cost vs. policy finetuning cost
>
> 1. **World model training steps.**
>    The world model is **not** trained for 150k steps; our previous description was inaccurate. The 150k steps shown in the figure actually refer to the *policy* training steps, not the world-model training steps. The world model is trained for 100k steps. We have clarifyied this in the paper in Table. 5 in Appendix.
>
> 2. **One world model for multiple tasks.**
>    Our current world model is indeed a *single* model that serves all four suites and 40 tasks, which is consistent with your expectation that the world model should be shared across multiple tasks.
>
> 3. **Whether to count world model training as “cost”.**
>
>    We believe that the training cost of the world model should not be directly counted as the per-task RL cost, because it is conceptually analogous to **building a simulator**: a one-time infrastructure effort rather than a per-task expense. Moreover, as research on world models progresses, they are likely to become increasingly large—similar to Cosmos-predict [2] or WoW [3]. For this reason, attributing the full cost of training the world model to each individual policy is indeed inappropriate.
>
> [2] Cosmos Predict 2.5 & Transfer 2.5: Evolving the World Foundation Models for Physical AI, arXiv (Oct 28)
>
> [3] WoW: Towards a World Omniscient World Model Through Embodied Interaction, arXiv

---

> > ### Comment · Reviewer_p7T1 · 2025-11-27
> >
> > Thanks to the author's reply, basically solved my problem. I maintain my positive score unchanged.

---

> > > ### Author Response · Authors · 2025-11-27
> > >
> > > Dear Reviewer p7T1,
> > >
> > > We are happy to hear that our response addressed your concerns well! Also, we appreciate your support for our work. If you have any further questions or suggestions, please do not hesitate to let us know.
> > >
> > > Best regards,
> > >
> > > Authors

---

### Official Review · Reviewer_Escc · 2025-10-31

**Soundness:** 3
**Presentation:** 3
**Contribution:** 3
**Rating:** 6
**Confidence:** 4

**Summary:**

Reinforcement learning from verifiable rewards (RLVR) was recently shown to provide substantial performance benefits when used in post-training of language-based reasoning models and agentic AI systems. Recent works (e.g. Guo et al., Improving vision-language-action model with online reinforcement learning, 2025) have extended this paradigm to vision-language-action (VLA) models for robotic manipulation, but unlike language-only models that generate their own rollouts via autoregressive text generation, training VLA models for robotics using online RL requires interaction with an environment external to the policy itself (e.g. a simulator or real hardware) which can significantly complicate the training pipeline. Instead, this work proposes to train a world model (action-conditioned autoregressive video prediction model) on data from the target environment and then subsequently use the trained world model as environment for policy RLVR. The authors derive a "verifiable" reward function from the world model based on video generation results (pixel-level + perception score) and demonstrate that VLA models finetuned using the world model require less real data, are more robust to environment perturbations, and have a qualitatively different action distribution than the base model on tasks from LIBERO (simulation benchmark).

**Strengths:**

This paper is generally well written, timely, technically sound. Concretely:

- **Originality:** I believe that this paper tackles an interesting and timely problem, and is likely to be of interest to the community. It provides a rather concise but well-rounded overview of related work including RLVR with LLMs, VLMs, and most recently VLAs. Although using a learned world model for RL has been explored extensively in model-based RL literature, I am not aware of any other papers that have yet extended the paradigm to VLAs and I would thus consider it to be an original and meaningful contribution to the field.
- **Writing:** The paper is generally well written and easy to follow, and the illustrations are helpful for understanding the technical contributions. I expect a reader familiar with large models and/or RL literature to be able to appreciate the technical details.
- **Extensive perturbation experiments:** I appreciate the thorough evaluation of policy robustness wrt environment perturbations. These are hardly surprising results for anyone familiar with RL, but it is great to see it validated empirically in the context of VLAs and learned world models. Experiments are clearly motivated and the discussion and analysis of results is informative.

**Weaknesses:**

While my assessment of the paper is positive overall, I do believe that there is room for improvement both in terms of writing as well as experiments:

- **Writing:** While the paper is generally easy to follow, it does appear rather rushed with frequent typographical and grammatical errors throughout. For example, some letters in the title are capitalized seemingly at random, and there are typos (*e.g.* L314 "We report success rate (SR) for all tasks**..** 3) Base Policy"). The paper would benefit from a round of proof-reading. Additionally, I would suggest the authors to be more explicit in their writing; instead of defining reward functions as "Reward type 1", "Reward type 2" etc. in the main text and then referencing them by their # in Table 4 and the accompanying illustration, it would be easier to follow if you referenced them via more descriptive names. Similarly, the base models used as baselines are introduced as Base (3w) and Base (15w) but it is never explicitly stated what 3w and 15w refers to.
- **Limited ablations:** The current set of experiments clearly demonstrate that RLVR with the trained world model improves policy performance, and the authors do conduct an ablation on the reward design which is informative. However, given that the world model is a key contribution of this work, I expected to see more analysis on the design and usage of the world model itself. For example, it is currently not clear how world model size (in terms of parameters) impacts generation quality nor policy RLVR, and it is also not clear what the relationship between RLVR iterations and policy performance is (the policy is only evaluated at 400 iterations). Experiments like these seem rather important given the nature of the problem, and I do not believe that they would be prohibitively expensive to run (policy evaluation at steps 100, 200, 300, for example, would be cheap if you already have the checkpoints, and you could similarly compare generation quality of the current world model size to that of a smaller world model).
- **World model training data:** It is not entirely clear to me what data is used to train the world model. If the world model is trained only on demonstrations (it is my understanding that this is the case), then I wouldn't expect the world model to generalize well to out-of-distribution states and actions for the same reasons that the policy does not. I would appreciate it if the authors could clarify this part and potentially back up any claims with data or references to prior work that addresses my concern.

**Questions:**

I would really appreciate it if the authors can address my comments in the "weaknesses" section above using written arguments and potentially additional experimental results. I would also appreciate it if the authors can commit to improving the writing beyond just correcting the specific instances I pointed out above.

---

> ### Author Response · Authors · 2025-11-26
> **4. Response to the Reviewer Escc**
>
> We thank the reviewer for the thoughtful and constructive feedback, and for the positive assessment of our work’s originality, technical soundness, and empirical robustness. Below we address each concern in detail and describe the changes we will make to the paper.
>
> ##### 4.1. **Writting**
>
> Thank you for your suggestion. We have revised the paper accordingly in response to your comments, and we will carefully address all remaining writing and presentation issues in the final version.
>
> ##### 4.2 **Limited ablations**
>
> We agree that it is important to show how policy performance evolves over the course of RLVR, rather than only at the final iteration.
>
> **RLVR iterations vs. policy performance (new added).**
> We have now evaluated the policy at intermediate RLVR steps for LIBERO Sptial tasks. Specifically, we evaluate the policy at RLVR iterations using the same evaluation protocol as in the main results. Notably, to better characterize the results, we now report outcomes under 3 seeds, providing a more comprehensive view of the performance.
>
> | Libero Spatial                 | Iter      | SR             | Avg   | Std  |
> | ------------------------------ | --------- | -------------- | ----- | ---- |
> | BasePolicy （Report in Paper） | 150k(sft) | 88.4           |       |      |
> | RFT（Report in Paper）         | 400(rft)  | 94.4           |       |      |
> | RFT                            | 200(rft)  | 94.4/94.0/94.8 | 94.4  | 0.40 |
> | RFT                            | 400(rft)  | 94.8/94.4/95.2 | 94.8  | 0.30 |
> | RFT                            | 600(rft)  | 94.4/95.2/95.2 | 94.93 | 0.50 |
>
> We believe these results further support the reviewer’s intuition: RLVR is effective early and continues to provide improvements, but with diminishing returns as the policy converges under the world-model-defined reward.
>
> **World model size vs. generation quality and RLVR performance (new added).**
>
> We conducted an additional ablation comparing our default world model with a larger variant, where we increased both the number of layers and the hidden dimension, resulting in roughly a 3× increase in parameters. We observe that as the world model grows larger, performance indeed improves further. This suggests that a more accurate world model provides better reward feedback. We plan to continue exploring the benefits of even stronger world models in future work.
>
> | Libero Spatial         | Iter     | WM Param Size | SR   |
> | ---------------------- | -------- | ------------- | ---- |
> | RFT（Report in Paper） | 400(rft) | 138m          | 94.4 |
> | RFT                    | 400(rft) | 421.15m       | 95.2 |
>
> #### 4.3. **World model training data**
>
> In the original setup, the world model was indeed trained only on expert data, without incorporating any additional data. To verify this point, we modified the training regime: instead of training the world model solely on expert trajectories from LIBERO-10, Spatial, Object, and Goal, we now train it on both the expert trajectories from LIBERO-10, Spatial, Object, and Goal **and** on sub-optimal trajectories from LIBERO-90. The results are shown in the following experiment:
>
> We indeed trained the world model based on demonstration data. In addition, we have now included an extra set of experiments that leverage failure data.
>
> | Libero Spatial           | Iter     | Extra Data | SR   |
> | ------------------------ | -------- | ---------- | ---- |
> | RFT（Report in Paper）   | 400(rft) | No         | 94.4 |
> | RFT (extra_failure_data) | 400(rft) | Yes        | 94.8 |

---

### Official Review · Reviewer_5Smb · 2025-11-01

**Soundness:** 2
**Presentation:** 3
**Contribution:** 2
**Rating:** 2
**Confidence:** 4

**Summary:**

The authors propose a method for improving Vision-Language-Action (VLA) models beyond imitation learning, by using a world model. Specifically, a two-stage pipeline is used: (Stage I) a learned (visual) world model is trained on offline data to predict the next image frame given the image history and an action; the policy is trained on the expert demonstration dataset; (Stage II) the policy is rolled out in the world model and the "verified reward" is given by perceptual similarity to the offline trajectories; this reward is used to optimize the policy using GRPO.

Experimentally: the authors demonstrate the world model can generate images similar to that from the offline dataset. On the LIBERO robotic benchmark, the method demonstrates several percentage point improvement for VLAs given 400 RL fine-tuning steps in the world model, demonstrates improvements over baseline given perturbations in the environment (backed up by visualization the resulting broader action distribution), and show improvement over simpler reward types.

**Strengths:**

The paper is clear and well-motivated; in particular, both world models and reinforcement learning as they pertain to VLAs is a critical, timely research topic which is likely to have significant impact. The methods section is well-exposited and the experiments are generally clear.

**Weaknesses:**

- Running RL in a world model is not a particularly novel idea; in fact, this is essentially the field of Model-Based Reinforcement Learning (MBRL). In the Related Work section, the authors differentiate their work from previous efforts by claiming the world model "also provides verified rewards". It is clear in many works (as a random example, [1]) that the world models can learn rewards as well; defining precisely the novelty here would be very helpful.
- Regarding the reward, it is unclear why perceptual distance to offline dataset trajectories is a "verified" reward. As the authors say in their conclusion, this design choice would constrain policies by offline dataset quality and limits the discovery of strategies beyond expert performance, which is a primary motivation behind the use of reinforcement learning in the first place.
- The actual improvement over baselines does not seem very significant: on the order of low single-digit percentage point improvements on LIBERO. It would be helpful to have standard error numbers to contextualize the improvement within variance.

[1] Janner, Michael, et al. "When to trust your model: Model-based policy optimization." Advances in neural information processing systems 32 (2019).

**Questions:**

- Standard error / error bars for experiment results would be very helpful to contextualize the improvement.
- Regarding Table 1, could you clarify whether the metrics are computed on the training data? It is natural that MSE would be low if the model is already trained; the more important test of the world model is on out-of-distribution states.
- Can you confirm which reward type was used in the main results?
- Re: Figure 4, is there some analysis (e.g. possibly with a quantitative metric) why having a broader action distribution makes sense as a result of the proposed training procedure, and why it helps for robustness?
- Could you expand on the choice of GRPO as the RL algorithm?

---

> ### Author Response · Authors · 2025-11-26
> **3. Response to the Reviewer 5Smb (part1)**
>
> We thank the reviewer for the thoughtful and constructive feedback. Before addressing your specific questions, we kindly ask you to first read our response to all reviewers, where we more clearly explain our work and clarify several misunderstandings.
>
> In summary, the core motivation and contribution of this paper is **not** simply to propose an algorithm that combines a VLA with a world model. Rather, our main goal is to open up a **new, third, world-model-centered paradigm for tackling the last-mile real-world deployment problem for VLAs**. This problem should be understood as an RL post-training problem in a non-ideal environment (where accurate rewards are unavailable, exploration is limited, and the exploration space is constrained). Therefore, viewing our work purely from the perspective of RL in idealized settings (e.g., simulation environments) naturally leads to many concerns and questions—which we have also summarized in the Limitations section.
>
> Below, we will address each concern in detail.
>
> #### 3.1. About the Novelty and the definition of verified rewards
>
> ##### 3.1.1 About the Novelty
>
> As discussed in our “General Response to All Reviewers,” this work is driven by a more fundamental, practice-oriented challenge: the **“last-mile” problem of deploying VLA policies in the real world**. A growing body of systems (e.g., **RL-100**, **DyNa**, **Sunday Robotics**) explicitly target policies that can (i) operate in **real environments for 24+ hours with near-100% reliability**, and (ii) be **continuously improved through real-world interaction**, rather than only via simulation.
>
> However, **existing dominant paradigms are insufficient to meet this last-mile requirement**:
>
> 1. **Pure real-world RL** (“real-robot RL”) faces critical limitations:
>    - Resetting and diversifying real scenes is extremely costly and heavily dependent on human labor.
>    - Many realistic tasks are **destructive, irreversible, or safety-critical** (e.g., grasping fragile objects, operations under wear-and-tear), which in practice forces current real-world RL experiments to focus on **“safe but relatively simple”** settings such as folding towels or plugging cables—far from reflecting the full complexity, risk profile, and long-horizon nature of real deployments.
> 2. **Purely simulation-based RL + sim-to-real** also has intrinsic limitations:
>    - Everyday interactions such as **wiping text off a whiteboard**, **complex material deformation**, **subtle friction and contact phenomena**, or **stochastic damage and wear** are notoriously difficult to capture faithfully in current simulators.
>    - For many task families, this leads to a substantial and often hard-to-bridge **sim-to-real gap**, especially when tasks require both high precision and long-term reliability.
>
> Consequently, we argue that **neither conventional real-world RL nor purely simulation-based approaches, in isolation, provide a scalable solution to the last-mile deployment problem for VLAs**.
>
> Against this backdrop, our work explores a **third, world-model-centered paradigm**:
>
> - We **learn a world model directly from real videos and real-world interactions**, thereby preserving rich, high-dimensional dynamics that traditional simulators struggle to model (including contact, deformation, and other non-ideal effects).
> - We then conduct large-scale **policy optimization inside this learned world model**, substantially improving **data efficiency** and relieving **reset, safety, and cost constraints** in the physical environment.
>
> Framed in this way, the **primary contribution of our work is not a specific new RL algorithmic trick**, but rather:
>
> 1. **Formulating and systematically analyzing a world-model-based paradigm for improving VLA policies in the real world**, explicitly targeting the **last-mile deployment** challenge (long-duration, high-reliability, continuously improving policies).
> 2. **Instantiating and empirically validating** this paradigm with a concrete world-model architecture and optimization pipeline, and rigorously **quantifying its benefits and limitations** along dimensions such as data efficiency, robustness to real-world variability, and the complexity of tasks that can be reliably handled.
>
> By the way, as noted by Reviewer **Escc**, our work is among the first to explicitly Extending the MBRL paradigm to VLAs in real-robot settings .
>
> > “Although using a learned world model for RL has been explored extensively in the model-based RL literature, I am not aware of any other papers that have yet extended the paradigm to VLAs and I would thus consider it to be an original and meaningful contribution to the field.”

---

> ### Author Response · Authors · 2025-11-26
> **3. Response to the Reviewer 5Smb (part2)**
>
> ##### 3.1.2 About the clarification of reward design
>
> "Verified rewards" is not "Learned rewards from a model". Many prior world-model-based works indeed learn a reward model jointly with the dynamics model. However, such rewards are not guaranteed to be correct,and may exhibit reward hacking or OOD failure. This type of reward is fundamentally not verifiable, the model may hallucinate success, misjudge safety, or become biased by the training distribution. Our verified reward is deterministically computable, not learned. In contrast, our reward is fully deterministic, task-independent, and does not require training. It is directly computed from pixel-space similarity between expert trajectories and world-model rollouts. This reward does not rely on any additional network or learned classifier. Thus, it is verifiable and robust against reward hacking and safe for scaling, similar to rule-based reward functions in RLHF for LLMs. Importantly, no existing VLA post-training work uses such verified pixel-space rewards for policy optimization. A corresponding clarification has also been incorporated into the **Related Work** section.

---

> ### Author Response · Authors · 2025-11-26
> **3. Response to the Reviewer 5Smb (part3)**
>
> #### 3.2. Why is perceptual distance a “verified” reward, and what about limiting beyond-expert strategies?
>
> First, we would like to reiterate that the challenges associated with using **ground-truth action-induced signals** are *not* incidental design flaws, but are instead inherent to the overarching goal articulated in our **“General Response to All Reviewers”**—namely, the pursuit of a **third, world-model-centered paradigm for solving real-world RL**.
>
> We are fully aware of the limitations introduced by this design choice, and we explicitly acknowledge and analyze them in the *Limitations* section. In our view, this reflects a **practical trade-off that any current attempt to deploy world models for real-robot RL must face** [2,3]. In many realistic real-world settings, practitioners do not have access to clean, simulator-style reward signals, and obtaining such signals can be prohibitively expensive or infeasible.
>
> Accordingly, our method is **not** intended to serve as an idealized solution to the problem of “optimizing VLA policies with a world model under perfect reward supervision.” Rather, it is deliberately positioned as a **pragmatic, transitional, and empirically effective compromise** for scenarios in which *reliable rewards are unavailable*, yet one still wishes to harness a world model to enable learning in real-robot RL.
>
> With this premise clarified, we now respond to your three questions in detail:
>
> ##### 3.2.1 About the "verified" reward
>
> First, we have provided a clearer and more explicit illustration of the “verified” reward in the 3.1.2. Next, we explain, at a fundamental level, why our definition of the “verified” reward is reasonable and well-justified.
>
> As we clarified in the **“General Response to All Reviewers”**, our goal is to use the world model as a *replacement* for both the simulator and the real-robot environment during training—i.e., as a cheap yet more realistic environment that can provide feedback signals for optimizing VLA policies.
>
> In this setting, unlike in standard simulation-based RL, we **cannot** obtain a simulator-style **binary reward for success or failure**. The world model does not expose a verifiable success flag, so we must construct rewards via additional mechanisms.
>
> There are two straightforward options:
>
> - (a) **Learn a reward model**, as in [4, 5];
> - (b) **Use ground-truth subgoal images to compute a verified reward**, as in [1, 2, 3].
>
> In this paper, we adopt **option (b)**. (The limitation of the (a) will be illustrated in 3.2.2 part.) Although the resulting reward is computed in an *indirect* way, it is still effective. This is consistent with many VLA approaches that rely on **video generation models**, where one typically generates future goal images and then uses an inverse dynamics model (IDM) to infer the corresponding actions. Under this design, the notion of task completion in **control space** and in **pixel space** becomes effectively aligned. The mechanism is indirect, but that does **not** preclude it from providing a meaningful or useful reward signal.
>
> Moreover, recent works such as **VLA-R1** [6] and **ThinkAct** [7] further support the idea that **consistency in pixel space** is often the most direct indicator of task execution quality. The main open question is not whether pixel space is appropriate, but rather **which pixel-space metric** to use—this is ultimately a modeling and design choice.
>
> By the way, our solution is also endorsed by Reviewer **p7T1**:
>
> > The data-driven world model is used as a verifiable reward generator instead of a traditional physical simulator, which skillfully avoids the problems of manual modeling bias and high cost. By calculating dense rewards between the generated and expert trajectories, a stable, aligned action-monitoring signal is provided for policy optimization, significantly reducing sample complexity.”
>
> [1] ReinboT: Amplifying Robot Visual-Language Manipulation with Reinforcement Learning. ICML2025
>
> [2] NORA-1.5: A Vision-Language-Action Model Trained using World Model- and Action-based Preference Rewards, arXiv (Nov 18)
> [3] SRPO: Self-Referential Policy Optimization for Vision-Language-Action Models, arXiv (Nov 19)
>
> [4] π\*0.6 : a VLA That Learns From Experience
>
> [5] World-Env: Leveraging World Model as a Virtual Environment for VLA Post-Training
>
> [6] VLA-R1: Enhancing Reasoning in Vision-Language-Action Models
>
> [7] ThinkAct: Vision-Language-Action Reasoning via Reinforced Visual Latent Planning

---

> ### Author Response · Authors · 2025-11-26
> **3. Response to the Reviewer 5Smb (part4)**
>
> ##### 3.2.2 About the limiting beyond-expert strategies
>
> We agree that our solution is still not optimal and we have already acknowledged this limitation in the paper (L478–480). Rather, they are a **pragmatic choice at the current stage**.
>
> For real-robot RL at the current stage, **task success rate** is often more important than whether the explored trajectory is the shortest or fastest possible. Under this perspective, using ground-truth action-induced trajectories is a simple yet effective way to make real-robot RL actually work in practice. This choice is also consistent with many recent works (e.g., SRPO [3], NORA-1.5 [2]), which optimize policies with respect to goal images or latent representations (e.g., JEPA features) derived from expert behaviors rather than from truly optimal trajectories.
>
> In contrast, designing a reward function/model is sometimes unreliable. Many recent approaches that attach a reward head to a VLM, or directly use a VLM as a verifier, are **highly dependent on the VLM’s understanding capability**. However, current VLMs are primarily trained for generic vision–language tasks and remain relatively weak in **robotics-specific spatial and dynamical reasoning**. Moreover, even for those works that perform task-specific finetuning, the underlying VLM models are typically not large enough to close this gap. As a result, the capability of current VLMs is still insufficient, and reward models built on top of them are prone to **reward hacking**.
>
> From our perspective, these are **different solution routes**, rather than strictly ordered, superior–inferior options. Using ground-truth action-induced trajectories is a feasible, though not ultimately optimal, approach at the current stage, and we look forward to more sophisticated solutions that can address these challenges in the future.
>
> #### 3.3 Magnitude of improvement and variance
>
> We understand the concern that the *absolute* improvement in success rate appears modest.
>
> 1. First, the VLA work we reference does not provide standard error numbers to contextualize the improvement relative to variance. However, we have conducted additional experiments here to address your question.
>
>    | Libero Spatial                 | Iter      | SR             | Avg   | Std  |
>    | ------------------------------ | --------- | -------------- | ----- | ---- |
>    | BasePolicy （Report in Paper） | 150k(sft) | 88.4           |       |      |
>    | RFT（Report in Paper）         | 400(rft)  | 94.4           |       |      |
>    | RFT                            | 200(rft)  | 94.4/94.0/94.8 | 94.4  | 0.40 |
>    | RFT                            | 400(rft)  | 94.8/94.4/95.2 | 94.8  | 0.30 |
>    | RFT                            | 600(rft)  | 94.4/95.2/95.2 | 94.93 | 0.50 |
>
> 2. As shown in Table 10 of the Appendix, the performance gains we achieve are already quite substantial compared to several other works in this area. Since the VLA model can reach a strong baseline performance with SFT alone, it is expected that the absolute improvement margin is relatively limited.
>
> 3. Moreover, compared with alternative approaches, our RFT procedure requires only **1% of their RL fine-tuning steps**, which is a particularly notable advantage given that we achieve comparable performance.

---

> ### Author Response · Authors · 2025-11-26
> **3. Response to the Reviewer 5Smb (part5)**
>
> #### 3.4 Questions
>
> ##### Q1. Standard error / error bars
>
> - We have provided the result in 3.3 paprt.
>
> ##### Q2. Table 1: train vs test metrics and OOD evaluation
>
> Thank you for pointing this out; our description was not sufficiently clear. We have revised on 4.2 Results Analysis. The metrics reported in **Table 1** are computed on the **test set**, not the training set.
>
> ##### Q3. Reward type used in main results
>
> In all main results, we use the **"Model-Based Image Consistency Reward "reward** defined in Table 4.
>
> The primary reason is that current world models are still not strong enough, and we want to ensure that the reward signal is not overly affected by imperfect image generation. Following the IMC (Internal Model Control) idea used in GEVRM[8], we compute the reward based on the images generated from the ground-truth actions and those generated from the predicted actions. This design helps mitigate the impact of potential limitations in the world model’s capability.
>
> [8] GEVRM: Goal-Expressive Video Generation Model For Robust Visual Manipulation, ICLR25
>
> ##### Q4. Figure 4: broader action distribution and robustness
>
> This advantage stems from the intrinsic strengths of RL-based optimization.
>
> A similar observation has been made in [11] where the authors directly compare RL and SFT. Their analysis shows that:  Expert/SFT actions tend to concentrate around the center of the action space, often reflecting repeated motion patterns and limited behavioral diversity. RL-trained policies produce actions that are more uniformly distributed across the action space, enabling the policy to cover a wider variety of states. As a result, policies trained with RL exhibit substantially stronger robustness than those trained purely via SFT.
>
> This aligns exactly with our findings. In our setting, RFT similarly encourages the policy to explore local variations around expert behaviors, resulting in a broader and smoother action distribution, as shown in Figure 4. This broader distribution is a well-established indicator of improved robustness: a policy that covers more of the action space is more capable of handling perturbations, unexpected states, and small deviations from the demonstrations.
>
> ##### Q5. expand on the choice of GRPO as the RL algorithm？
>
> GRPO provides a more stable, lightweight, and large-model–friendly alternative to PPO. It offers direct reward optimization with lower computational overhead and better robustness, and subsequent work has already explored optimizing it with alternative algorithms (e.g., using DPO in SRPO [3]).

---

### Official Review · Reviewer_eHXZ · 2025-11-01

**Soundness:** 1
**Presentation:** 2
**Contribution:** 2
**Rating:** 2
**Confidence:** 4

**Summary:**

The paper proposes a framework for reinforcement fine-tuning in world simulators to train VLA models. It takes two stages: (1) it trains a world model (WM) on offline data and a base VLA policy with supervised finetuning (SFT); (2) it uses the learned WM to roll out actions while predicting visual trajectory. RL finetuning is done by computing a verifiable, trajectory-level reward that minimizes the difference between the ground-truth-action-induced rollouts. The proposed VLA-RFT improves performance and robustness with very few fine-tuning steps (~400) compared to SFT.

**Strengths:**

1. The paper is clearly written. The duet of the world model and the VLA model is formulated in a shared POMDP framework.

2. The efficiency claim is compelling. The paper reports several orders of magnitude fewer iterations required for RL fine-tuning compared to supervised baselines. This sample efficiency gain is attractive for practical post-training workflows.

**Weaknesses:**

1. Comparison with (Tan et al., 2025). Both papers share the same high-level idea of using "reinforcement Fine-tuning" and "verified rewards". However, the implementations are at least vastly different while not being exactly the opposite. First of all, (Tan et al., 2025) indeed leverages **interactive** feedback from the simulator. The claim from this submission (L074-075; L116-117) seems to imply that they don't, which is quite misleading. Second, (Tan et al., 2025) use a binary reward for success or failure, which is also **verifiable**.  In the meantime, the "verifiability" of this submission is less convincing in that (1) the verified reward is defined by a sum of reconstruction and similarity loss, which is much more indirect to the final task, and (2) the verified reward (or reconstruction loss) is computed against the trajectory induced by the ground-truth actions. There are three potential issues:

    (1) Are ground-truth action-induced trajectories optimal? Not quite, since many tasks are multi-modal, meaning that there are multiple possible trajectories leading to the same goal;

    (2) Are ground-truth action-induced trajectories easier to get? Not quite either, the simulator itself is cheaper while the world model is likely bigger (details are missing??);

    (3) Does RL bring explorativeness? Unfortunately, no again, since this relies on your SFT recording, which is a highly overlapped effort compared to the SFT phase. Meanwhile, the simulator is more likely to bring more diversity beyond the SFT recording data.

2. No real-world validation. The high-level idea is to learn a world model that emulates the simulator. This introduces a simulator-WM gap in addition to the already widely known real-sim gap. We need to answer the question: how large is this simulator-WM gap compared to the real-sim gap? The sim-to-real gap has been a big problem. The proposed method further complicated it and left many questions unanswered.

**Questions:**

Please see my questions in the weakness section.

---

> ### Author Response · Authors · 2025-11-26
> **2. Response to the Reviewer eHXZ (part1)**
>
> #### 2.1. As to RIPT‑VLA (Tan et al., 2025) in L74–75
>
> We would like to clarify that the remarks in L074–075 and L116–117 are intended as a direct citation of the conclusions of Tan et al. (2025), rather than a criticism or mischaracterization of their work. We share their view that training purely from offline datasets can induce distribution shift and related issues. Consistently, in L068–069, we also explicitly categorize Tan et al. (2025) as **simulation-based RL**, rather than **offline RL**.
>
> #### 2.2. As to the "verifiability" of reward design
>
> ##### 2.2.1 "indirect to the final task":
>
> As we clarified in the **“General Response to All Reviewers”**, our goal is to use the world model as a *replacement* for both the simulator and the real-robot environment during training—i.e., as a cheap yet more realistic environment that can provide feedback signals for optimizing VLA policies.
>
> In this setting, unlike in standard simulation-based RL, we **cannot** obtain a simulator-style **binary reward for success or failure**. The world model does not expose a verifiable success flag, so we must construct rewards via additional mechanisms.
>
> There are two straightforward options:
>
> - (a) **Learn a reward model**, as in [4, 5];
> - (b) **Use ground-truth subgoal images to compute a verified reward**, as in [1, 2, 3].
>
> In this paper, we adopt **option (b)**. Although the resulting reward is computed in an *indirect* way, it is still effective. This is consistent with many VLA approaches that rely on **video generation models**, where one typically generates future goal images and then uses an inverse dynamics model (IDM) to infer the corresponding actions. Under this design, the notion of task completion in **control space** and in **pixel space** becomes effectively aligned. The mechanism is indirect, but that does **not** preclude it from providing a meaningful or useful reward signal.
>
> Moreover, recent works such as **VLA-R1** [6] and **ThinkAct** [7] further support the idea that **consistency in pixel space** is often the most direct indicator of task execution quality. The main open question is not whether pixel space is appropriate, but rather **which pixel-space metric** to use—this is ultimately a modeling and design choice.
>
> [1] ReinboT: Amplifying Robot Visual-Language Manipulation with Reinforcement Learning. ICML2025
>
> [2] NORA-1.5: A Vision-Language-Action Model Trained using World Model- and Action-based Preference Rewards, arXiv (Nov 18)
> [3] SRPO: Self-Referential Policy Optimization for Vision-Language-Action Models, arXiv (Nov 19)
>
> [4] π\*0.6 : a VLA That Learns From Experience
>
> [5] World-Env: Leveraging World Model as a Virtual Environment for VLA Post-Training
>
> [6] VLA-R1: Enhancing Reasoning in Vision-Language-Action Models
>
> [7] ThinkAct: Vision-Language-Action Reasoning via Reinforced Visual Latent Planning
>
> By the way, our solution is also endorsed by Reviewer **p7T1**:
>
> > The data-driven world model is used as a verifiable reward generator instead of a traditional physical simulator, which skillfully avoids the problems of manual modeling bias and high cost. By calculating dense rewards between the generated and expert trajectories, a stable, aligned action-monitoring signal is provided for policy optimization, significantly reducing sample complexity.”

---

> ### Author Response · Authors · 2025-11-26
> **2. Response to the Reviewer eHXZ (part2)**
>
> ##### 2.2.2 "ground-truth action-induced":
>
> First, we would like to reiterate that the challenges associated with using **ground-truth action-induced signals** are *not* incidental design flaws, but are instead inherent to the overarching goal articulated in our **“General Response to All Reviewers”**—namely, the pursuit of a **third, world-model-centered paradigm for solving real-world RL**.
>
> We are fully aware of the limitations introduced by this design choice, and we explicitly acknowledge and analyze them in the *Limitations* section. In our view, this reflects a **practical trade-off that any current attempt to deploy world models for real-robot RL must face** [2,3]. In many realistic real-world settings, practitioners do not have access to clean, simulator-style reward signals, and obtaining such signals can be prohibitively expensive or infeasible.
>
> Accordingly, our method is **not** intended to serve as an idealized solution to the problem of “optimizing VLA policies with a world model under perfect reward supervision.” Rather, it is deliberately positioned as a **pragmatic, transitional, and empirically effective compromise** for scenarios in which *reliable rewards are unavailable*, yet one still wishes to harness a world model to enable learning in real-robot RL.
>
> With this premise clarified, we now respond to your three questions in detail:
>
> ---
>
> ##### Q1: Are ground-truth action-induced trajectories optimal?
>
> We agree that ground-truth action-induced trajectories are not strictly optimal, especially for multimodal tasks where multiple distinct trajectories can achieve the same goal. We have already acknowledged this limitation in the paper (L478–480).
>
> Our work does not claim that ground-truth trajectories are the best possible supervision signal. Rather, they are a **pragmatic choice at the current stage**.
>
> For real-robot RL at the current stage, **task success rate** is often more important than whether the explored trajectory is the shortest or fastest possible. Under this perspective, using ground-truth action-induced trajectories is a simple yet effective way to make real-robot RL actually work in practice. This choice is also consistent with many recent works (e.g., SRPO [3], NORA-1.5 [2]), which optimize policies with respect to goal images or latent representations (e.g., JEPA features) derived from expert behaviors rather than from truly optimal trajectories.
>
> In contrast, designing a reward function is sometimes unreliable. Many recent approaches that attach a reward head to a VLM, or directly use a VLM as a verifier, are **highly dependent on the VLM’s understanding capability**. However, current VLMs are primarily trained for generic vision–language tasks and remain relatively weak in **robotics-specific spatial and dynamical reasoning**. Moreover, even for those works that perform task-specific finetuning, the underlying VLM models are typically not large enough to close this gap. As a result, the capability of current VLMs is still insufficient, and reward models built on top of them are prone to **reward hacking**.
>
> From our perspective, these are **different solution routes**, rather than strictly ordered, superior–inferior options. Using ground-truth action-induced trajectories is a feasible, though not ultimately optimal, approach at the current stage, and we look forward to more sophisticated solutions that can address these challenges in the future.

---

> ### Author Response · Authors · 2025-11-26
> **2. Response to the Reviewer eHXZ (part3)**
>
> ##### Q2: Are ground-truth action-induced trajectories easier/cheaper to obtain than simulation?
>
> Returning to the motivation stated in our **“General Response to All Reviewers”**, we would like to reiterate that our goal is *not* to compete with RL in carefully engineered simulators. Our intention is to address the kinds of problems that **cannot** be adequately solved in simulation in the first place; from this perspective, a direct comparison with simulation-based RL is of limited relevance (similar to the real-robot experiments in Section 1.3). We temporarily use simulation **only** as a convenient testbed to verify that our overall approach is effective, because for real-robot manipulation of deformable objects, it is very difficult to construct a fair comparison against traditional methods.
>
> For many realistic scenarios, building a usable simulator is either prohibitively expensive or practically infeasible. For example, tasks such as **“wiping a whiteboard”** involve complex contact, deformable surfaces, and messy real-world conditions that are extremely hard to model accurately in conventional physics engines. In such cases, relying on simulation-based RL is often not viable in practice, despite its theoretical sample efficiency.
>
> By contrast, a **video-based world model** can be trained directly on real data collected from the physical environment and can implicitly capture rich visual and physical phenomena without requiring explicit manual modeling or domain-specific simulators. This is precisely the regime where we believe the **“third route”**—world-model-based RL between pure simulation RL and pure real-world RL—becomes most valuable.
>
> Finally, our world model is designed following iVideoGPT [6,8], which is relatively **lightweight and scalable** compared to very large video models. In our current experiments, both training and inference are of moderate cost and do not dominate the overall pipeline.
>
> [8] iVideoGPT: Interactive VideoGPTs are Scalable World Models, NIPS2024
>
> ---

---

> ### Author Response · Authors · 2025-11-26
> **2. Response to the Reviewer eHXZ (part4)**
>
> ##### Q3: Does RL in your framework bring explorativeness?
>
> We agree that, in the **traditional RL sense of active exploration**, our current framework does **not** provide strong explorativeness. As we emphasized in the **“General Response to All Reviewers”**, this is a **trade-off** at the current stage of our system design.
>
> Concretely, our setting combines:
>
> - a **strong pretrained VLA base model** with rich priors, and
> - a **world model** that has some ability to explore and to provide **verified rewards**.
>
> Under these assumptions, our goal in post-training is *not* to discover entirely new behaviors far away from the demonstration distribution. Instead, we want to use the world-model-generated reward to:
>
> - **penalize bad or unsafe actions**, and
> - more accurately complete tasks with a focus on robustness (ideally approaching “near-100%” reliability on the relevant distribution).
>
> From this perspective, VLA post-training does **not require large-scale active exploration** in the classical RL sense. The exploration we care about is *local*: staying within a neighborhood of the ground-truth behavior while exploring small variations, and this is precisely what the world model can support. In other words, the world model contributes diversity and coverage around the data manifold, while RL is used as a **targeted refinement mechanism** rather than as a broad exploration engine.
>
> Even under limited exploration, we believe our RL-based post-training remains **meaningful and complementary to SFT** for two reasons:
>
> 1. **Different optimization geometry compared to SFT.**
>    As discussed in [9], RLVR-style optimization and SFT exhibit fundamentally different optimization behaviors:
>
>    - SFT behaves like “climbing over the mountain”: it tends to modify the **principal directions** of the model’s parameters, which can distort the spectral structure of the pretrained model.
>    - RLVR behaves more like “taking a detour with a compass”: it avoids heavily modifying the principal directions and instead updates along **low-curvature paths**, thus better preserving the pretrained model’s spectral properties.
>
>    Thus, even if SFT and RL are trained on **overlapping trajectories**, the resulting models can still differ substantially in terms of robustness, calibration, and how they leverage the pretrained base model.
>
> 2. **Post-training for error avoidance rather than policy-from-scratch learning.**
>    In the context of VLA post-training, it is an **open question** whether we truly need full-fledged, exploration-heavy RL as in classic control/RL settings. Given a strong pretrained VLA base model, our view is that post-training should primarily:
>
>    - **suppress bad or unsafe actions**, and
>    - refine the model so that it more reliably avoids errors and behaves consistently,
>
>    rather than learning entirely new strategies from scratch. In this sense, RLVR acts as a **fine-grained refinement mechanism** on top of the base model, somewhat analogous to pruning or targeted adjustment that aims to **activate and stabilize** existing capabilities, rather than expanding coverage via aggressive exploration. This perspective is aligned with [10], which questions to what extent RL truly incentivizes reasoning “beyond” the base model versus refining it.
>
> Therefore, although our framework does not implement strong exploration in the classic RL sense, we argue that it still plays a valuable role as a **post-training method for VLA base models**, focusing on stability, error avoidance, and preservation of pretrained structure, while leveraging the world model to provide local diversity around the ground-truth distribution rather than broad exploratory coverage.
>
> [9] *The Path Not Taken: RLVR Provably Learns Off the Principals*, Arxiv
> [10] *Does Reinforcement Learning Really Incentivize Reasoning Capacity in LLMs Beyond the Base Model?* Arxiv

---

> ### Author Response · Authors · 2025-11-26
> **2. Response to the Reviewer eHXZ (part5)**
>
> #### 2.3. As to No real-world validation.
>
> We have also conducted real-world experiments, as presented in Section 1.1. Notably, on tasks that are difficult or even impossible to faithfully model in simulation, we **do not rely on any real-robot RL fine-tuning**, yet still obtain **highly competitive and visually compelling results**.
>
> Here, we would like to emphasize again that our goal in using a world model is **not** to compete with simulators, but to tackle tasks that simulators fundamentally struggle with. For example, chalk marks on a blackboard, or a wide variety of non-rigid objects with different materials. Expecting simulators to faithfully model all such real-world robotic scenarios would typically require extensive manual tuning and careful design of physical parameters.
>
> By contrast, these tasks are often more amenable to being modeled by **world models**. Recent large world models (e.g., WoW [9]) already demonstrate this potential. For instance, WoW can zero-shot generate a scenario where a lemon is dissolved by acid—something that is extremely difficult to realize in a conventional simulator. Enabling robots to experience and learn from precisely these kinds of scenarios, which are very hard to model in simulation, is where world models have a unique advantage.
>
> Of course, world models still have many limitations; we explicitly discuss these in L480–482. Nevertheless, we believe there is substantial room for further exploration and improvement. This is exactly the motivation and value of our work: to encourage the community to systematically study this relatively underexplored problem. This emerging trend is also reflected in several very recent works [11,12,13] that pursue similar directions. Together with ours, these works suggest that this line of research is promising and worth continued investigation.
>
> [11] **WoW: Towards a World Omniscient World Model Through Embodied Interaction**, arXiv
> [12] **WMPO: World Model-based Policy Optimization for Vision-Language-Action Models**, arXiv
> [13] **Reinforcing Action Policies by Prophesying**, arXiv

---

### Author Response · Authors · 2025-11-26
**1. General Response to All Reviewers (part1)**

We thank the reviewers for their thoughtful comments and for engaging deeply with our work. Several reviews primarily interpret our contribution through the lens of *world-model + VLA optimization in simulation*. While this is indeed one component of our method, it does not fully capture the central motivation and intended impact of our work. In this section, we clarify the problem we aim to address and the paradigm we propose, which we hope will help align our work with the reviewers’ expectations.

#### 1.1 Clarifications of Our Motivation

Our work is driven by a more fundamental, practice-oriented challenge: the **“last-mile” problem of deploying VLA policies in the real world**. A growing body of systems (e.g., **RL-100**, **DyNa**, **Sunday Robotics**) explicitly target policies that can (i) operate in **real environments for 24+ hours with near-100% reliability**, and (ii) be **continuously improved through real-world interaction**, rather than only via offline data or simulation.

However, **existing dominant paradigms are insufficient to meet this last-mile requirement**:

1. **Pure real-world RL** (“real-robot RL”) faces critical limitations:
   - Resetting and diversifying real scenes is extremely costly and heavily dependent on human labor.
   - Many realistic tasks are **destructive, irreversible, or safety-critical** (e.g., grasping fragile objects, operations under wear-and-tear), which in practice forces current real-world RL experiments to focus on **“safe but relatively simple”** settings such as folding towels or plugging cables—far from reflecting the full complexity, risk profile, and long-horizon nature of real deployments.
2. **Purely simulation-based RL + sim-to-real** also has intrinsic limitations:
   - Everyday interactions such as **wiping text off a whiteboard**, **complex material deformation**, **subtle friction and contact phenomena**, or **stochastic damage and wear** are notoriously difficult to capture faithfully in current simulators.
   - For many task families, this leads to a substantial and often hard-to-bridge **sim-to-real gap**, especially when tasks require both high precision and long-term reliability.

Consequently, we argue that **neither conventional real-world RL nor purely simulation-based approaches, in isolation, provide a scalable solution to the last-mile deployment problem for VLAs**.

Against this backdrop, our work explores a **third, world-model-centered paradigm**:

- We **learn a world model directly from real videos and real-world interactions**, thereby preserving rich, high-dimensional dynamics that traditional simulators struggle to model (including contact, deformation, and other non-ideal effects).
- We then conduct large-scale **policy optimization inside this learned world model**, substantially improving **data efficiency** and relieving **reset, safety, and cost constraints** in the physical environment.

Framed in this way, the **primary contribution of our work is not a specific new RL algorithmic trick**, but rather:

1. **Formulating and systematically analyzing a world-model-based paradigm for improving VLA policies in the real world**, explicitly targeting the **last-mile deployment** challenge (long-duration, high-reliability, continuously improving policies).
2. **Instantiating and empirically validating** this paradigm with a concrete world-model architecture and optimization pipeline, and rigorously **quantifying its benefits and limitations** along dimensions such as data efficiency, robustness to real-world variability, and the complexity of tasks that can be reliably handled.

We view this paradigm as complementary to, rather than a replacement for, existing real-robot and simulation-based methods; its value lies in **bridging the gap between scalable policy optimization and realistic real-world dynamics**.

Finally, as noted by Reviewer **Escc**, our work is among the first to explicitly pursue this world-model-based route for VLA deployment. The subsequent emergence of several closely related efforts (e.g., **WMPO** [1], **NORA-1.5** [2], **SRPO** [3], and **Reinforcing Action Policies by Prophesying** [4]), which adopt similar world-model-centered ideas for VLA training and optimization, further underscores both the **timeliness** and the **importance** of the problem we target. We believe this contemporaneous follow-up work provides additional evidence that the paradigm we advocate addresses a meaningful and emerging need in the community.

[1] WMPO: World Model-based Policy Optimization for Vision-Language-Action Models, arXiv (Nov 2)
 [2] NORA-1.5: A Vision-Language-Action Model Trained using World Model- and Action-based Preference Rewards, arXiv (Nov 18)
 [3] SRPO: Self-Referential Policy Optimization for Vision-Language-Action Models, arXiv (Nov 19)
 [4] Reinforcing Action Policies by Prophesying, arXiv (Nov 25)

---

### Author Response · Authors · 2025-11-26
**1. General Response to All Reviewers (part2)**

#### 1.2 Clarifications of Our Contribution

As discussed in Section 1.1, the broader significance of our paradigm lies in its applicability to **real-robot deployment**. Our initial submission indeed did **not** include real-robot experiments, which is a major limitation; we will address this by adding real-robot results in Section 1.3 of the revised version.

That said, we would like to emphasize that **even setting aside real-robot experiments, our work already has substantial value and significance**, because *operationalizing* this paradigm required overcoming several non-trivial challenges:

1. **Before our work, there were almost no optimization algorithms tailored to flow-based VLAs.**
   Most existing RL methods for VLA-style policies were designed for *autoregressive (AR)* models (e.g., OpenVLA [6]), and only very recently have a few works such as π_{RL} [5] begun to explore RL fine-tuning for **flow-based** VLA models.
2. **There have been very few action-conditioned world models suitable for robotics.**
   Only very recent efforts such as *Cosmos Predict 2.5* [7] started to seriously investigate large-scale, action-conditioned world models that can capture rich physical dynamics for embodied agents.
3. **To the best of our knowledge, prior to our paper there was no work that discusses how to jointly optimize a VLA policy and a world model within a unified framework.**
   Our paper is, to our knowledge, the first to propose **how to leverage a learned world model to jointly optimize a VLA policy**, including the concrete training pipeline and design choices. Only *after* our submission did several concurrent works [1,2,3,4] appear, which further highlight the importance of this direction.

[5] π_{RL}: Online RL Fine-tuning for Flow-based Vision-Language-Action Models, arXiv (Oct 29)
[6] OpenVLA: An Open-Source Vision-Language-Action Model, arXiv
[7] Cosmos Predict 2.5 & Transfer 2.5: Evolving the World Foundation Models for Physical AI, arXiv (Oct 28)

#### 1.3 Clarifications of Our Real-World Experiments

To further demonstrate the practical value of our approach on real robots, we conducted the following real-world experiments:

1. **Experimental setup.**
   We consider a long-horizon, deformable-object manipulation task that is difficult to simulate: **folding a towel**. We collect 50 real-world trajectories using a single-arm robot that folds the towel, with **only a third-person camera** available (no wrist or depth cameras).

2. **Baselines and training details.**
   We use **VLA-Adapter** as the base policy model and train two supervised fine-tuning checkpoints with **20k** and **80k** steps, respectively. The world model is trained for **24k** steps, and we then post-train the policy with the world model for **200 RFT steps**. We evaluate all variants using **folding success rate** and **average task completion time** as metrics.

3. **Results.**
   We find that, after **20k** steps of supervised fine-tuning (SFT), **additional 60k SFT steps provide no improvement in success rate**. In contrast, applying our **RFT-based fine-tuning** on top of the 20k-SFT checkpoint boosts the **success rate to 100% (10/10)**. The improvement is particularly pronounced for failure modes such as failing to grasp the towel correctly or dropping it while moving after a successful grasp.

   In particular, at the critical moments of grasping a deformable object, the SFT-only policy often fails to grasp the towel reliably because of its highly variable shape, leading to a significantly reduced success rate. In contrast, after RFT, the policy becomes much more precise at grasping such deformable objects and reaches a **100% success rate** on this task.

   Although these experiments are still preliminary, they already demonstrate that in scenarios where simulation cannot faithfully model the environment for RL optimization, our paradigm substantially advances a **third, real-world–oriented RL paradigm** based on world models.

**Detailed trial results.**

| Real world | Iter                  | SR    | Time(1)    | 2    | 3          | 4    | 5        | 6           | 7        | 8    | 9        | 10   |
| ---------- | --------------------- | ----- | ---------- | ---- | ---------- | ---- | -------- | ----------- | -------- | ---- | -------- | ---- |
| BasePolicy | 20k (SFT)             | 05/10 | poor grasp | 1'00 | no grasp   | 0'56 | no grasp | joint limit | no grasp | 0'55 | 0'52     | 0'51 |
| BasePolicy | 80k (SFT)             | 05/10 | poor grasp | 0'48 | poor grasp | 0'42 | 0'50     | joint limit | no grasp | 0'48 | no grasp | 0'49 |
| RFT        | 20k (SFT) + 200 (RFT) | 10/10 | 0'41       | 0'54 | 1'00       | 0'54 | 0'56     | 0'50        | 0'52     | 0'50 | 1'00     | 1'02 |

---

### Author Response · Authors · 2025-12-02
**Rebuttal Summary**

Dear ACs,

We sincerely thank the reviewers for their constructive and insightful comments. We are encouraged that the reviewers recognized the novelty and effectiveness of our proposed framework, especially as the first work to explore world-model-based post-training for real-robot VLAs, providing timely guidance for the community (Reviewers [eHXZ, 5Smb, Escc]).

Although we had not completed real-robot experiments at the time of submission, we have now included real-robot validation (post-training for deformable object manipulation), achieving a 100% success rate (compared to 50% with SFT). This demonstrates that, relative to RL in simulation followed by sim-to-real transfer, our approach can tackle tasks that are difficult or impossible to handle purely in simulation. Compared to real-robot RL, the efficiency of our post-training (Reviewers [p7T1, eHXZ]) is also a distinctive advantage.

We believe our work is helping to establish a third paradigm: using world models as the core to address real-robot VLA RL post-training. Our work has already inspired a number of follow-up efforts [1,2,3,4]. We are also pleased that the reviewers recognized the importance of our contribution:

“I am not aware of any other papers that have yet extended the paradigm to VLAs and I would thus consider it to be an original and meaningful contribution to the field.”

[1] WMPO: World Model-based Policy Optimization for Vision-Language-Action Models, arXiv (Nov 2)
[2] NORA-1.5: A Vision-Language-Action Model Trained using World Model- and Action-based Preference Rewards, arXiv (Nov 18)
[3] SRPO: Self-Referential Policy Optimization for Vision-Language-Action Models, arXiv (Nov 19)
[4] Reinforcing Action Policies by Prophesying, arXiv (Nov 25)

We have carefully revised our paper, taking all suggestions into account. Below are our detailed responses to each point.

---

### Meta-Review · Area_Chair_ce5g · 2026-01-06

**Summary:**

The paper proposes VLA-RFT, using learned world models as simulators for reinforcement fine-tuning of Vision-Language-Action models. Reviewers eHXZ and 5Smb (both rating: 2) raised critical concerns about novelty, the "verified reward" design, and lack of real-world validation. Reviewer eHXZ noted that ground-truth action-induced trajectories are neither optimal nor easier to obtain than simulator-based approaches, and that the framework provides no meaningful exploration beyond demonstrations. Most critically, the complete absence of real-world validation is problematic given that the approach introduces a world-model gap on top of the existing sim-to-real gap. Reviewer 5Smb questioned why perceptual distance to offline trajectories constitutes a "verified" reward and noted that this design fundamentally limits policies to expert performance, contradicting a primary RL motivation. Experimental improvements were modest without proper error quantification. Reviewers Escc and p7T1 (both rating: 6) were more positive, acknowledging the extension of RLVR to VLAs as original, but noted limited ablations, unclear generalization properties, and that expert-similarity rewards cannot discover better strategies.

While the paper addresses a relevant problem, the fundamental limitations outweigh contributions. The framework lacks meaningful exploration, is limited to expert-level performance by design, introduces an additional modeling gap, and shows only modest empirical gains. The positioning as a "third paradigm" for real-world deployment is not adequately supported

**Reviewer Concerns:**

The authors provided extensive responses including preliminary real-world experiments on towel folding and reframed their contribution as establishing a "third paradigm" between simulation RL and real-world RL. However, the rebuttal did not resolve core concerns. The authors acknowledged that ground-truth trajectories are not optimal and that their framework lacks meaningful exploration, essentially validating reviewer concerns rather than addressing them. They framed these as "pragmatic trade-offs" rather than resolving the fundamental limitations. The real-world experiments are preliminary and came too late for proper evaluation. Variance estimates help contextualize the 4-6 percentage point gains but still represent incremental improvement given the added complexity.

**Reviewer Scores:**

Reviewers eHXZ and 5Smb would likely maintain their scores of 2, as fundamental issues with reward design, lack of exploration, and insufficient validation remain unresolved. Reviewers Escc and p7T1 would likely maintain scores around 5-6, remaining borderline given that core methodological limitations persist and real-world validation is inadequate.

---

### Decision · Program_Chairs · 2026-01-26

Reject